# Flash-GRPO: Efficient Alignment for Video Diffusion via One-Step Policy Optimization

Xiaoxuan He [1 2 * †]  Siming Fu [2 * ‡]  Zeyue Xue [2]  Weijie Wang [1]  Ruizhe He [2]  Yuming Li [2]  Dacheng Yin [3]
Shuai Dong [2]  Haoyang Huang [2]  Hongfa Wang [4]  Nan Duan [2]  Bohan Zhuang [1 ‡]

## Abstract

Group Relative Policy Optimization has emerged as essential for aligning video diffusion models with human preferences, but faces a critical computational bottleneck: training a 14B parametered model typically demands hundreds of GPU days per experiment. Existing efficiency methods reduce costs through sliding window subsampling training timesteps, but fundamentally compromise optimization, exhibiting severe instability and failing to reach full trajectory performance. We present **Flash-GRPO**, a single-step training framework that outperforms full trajectory training in alignment quality under low computational budgets while substantially improving training efficiency. Flash-GRPO addresses two critical challenges: *iso-temporal grouping* eliminates timestep-confounded variance by enforcing prompt-wise temporal consistency, decoupling policy performance from timestep difficulty; *temporal gradient rectification* neutralizes the time-dependent scaling factor that causes vastly inconsistent gradient magnitudes across timesteps. Experiments on 1.3B to 14B parameter models validate Flash-GRPO's effectiveness, demonstrating substantial training acceleration with consistent stability and state-of-the-art alignment quality.

## 1. Introduction

Video diffusion models (Ho et al., 2022; Blattmann et al., 2023; Hong et al., 2022; Gao et al., 2025) have achieved remarkable progress in generating realistic and temporally consistent videos. However, aligning these models with human preferences such as aesthetic quality, prompt adherence, and physical plausibility remains a critical challenge. Reinforcement Learning (RL) has emerged as the dominant paradigm for this alignment task (Shao et al., 2024; Zheng et al., 2025; Yu et al., 2025; Zhao et al., 2025), with recent methods like Flow-GRPO (Liu et al., 2025a) and Dance-GRPO (Xue et al., 2025) successfully adapting Group Relative Policy Optimization (GRPO) to video generation, demonstrating substantial improvements in generation quality.

Despite these advances, a fundamental computational barrier persists: video diffusion models must backpropagate gradients through spatiotemporal latents across long denoising trajectories. Standard GRPO approaches require computing gradients over the *full trajectory* for every timestep. This dense supervision creates prohibitive memory consumption and severely limits training throughput. As illustrated in Figure 1, aligning a 14B parameter video model typically demands hundreds of GPU days per experiment, imposing a scalability bottleneck that restricts both research iteration and practical deployment.

Existing efficiency methods such as Flow-GRPO-Fast (Liu et al., 2025a) and MixGRPO (Li et al., 2025) attempt to reduce this cost through sliding window subsampling, training on only a small subset of consecutive timesteps. While this reduces computation, our analysis reveals a fundamental flaw: naive subsampling compromises the optimization landscape. As shown in Figure 2, **one-step version** exhibits severe training instability and fails to reach the performance ceiling of full-trajectory training, creating an undesirable trade-off between efficiency and quality. The core issue is twofold: first, mixing timesteps within advantage groups introduces confounded variance that obscures the true policy signal; second, time-dependent gradient scaling factors cause different timesteps to contribute inconsistently to parameter updates, destabilizing optimization. *This raises a natural question: can we design a single-step training paradigm that matches full trajectory performance while maximizing computational efficiency?*

In this work, we present **Flash-GRPO**, a single-step training

---

[*]Equal contribution. [†] Work was done during an internship in Joy Future Academy. [‡] Corresponding authors. [1]Zhejiang University [2]Joy Future Academy [3]Independent Researcher [4]Tsinghua University. Correspondence to: Siming Fu <fusiming.chosen@jd.com>, Bohan Zhuang <bohan.zhuang@zju.edu.cn>.

*Proceedings of the 43rd International Conference on Machine Learning*, Seoul, South Korea. PMLR 306, 2026. Copyright 2026 by the author(s).

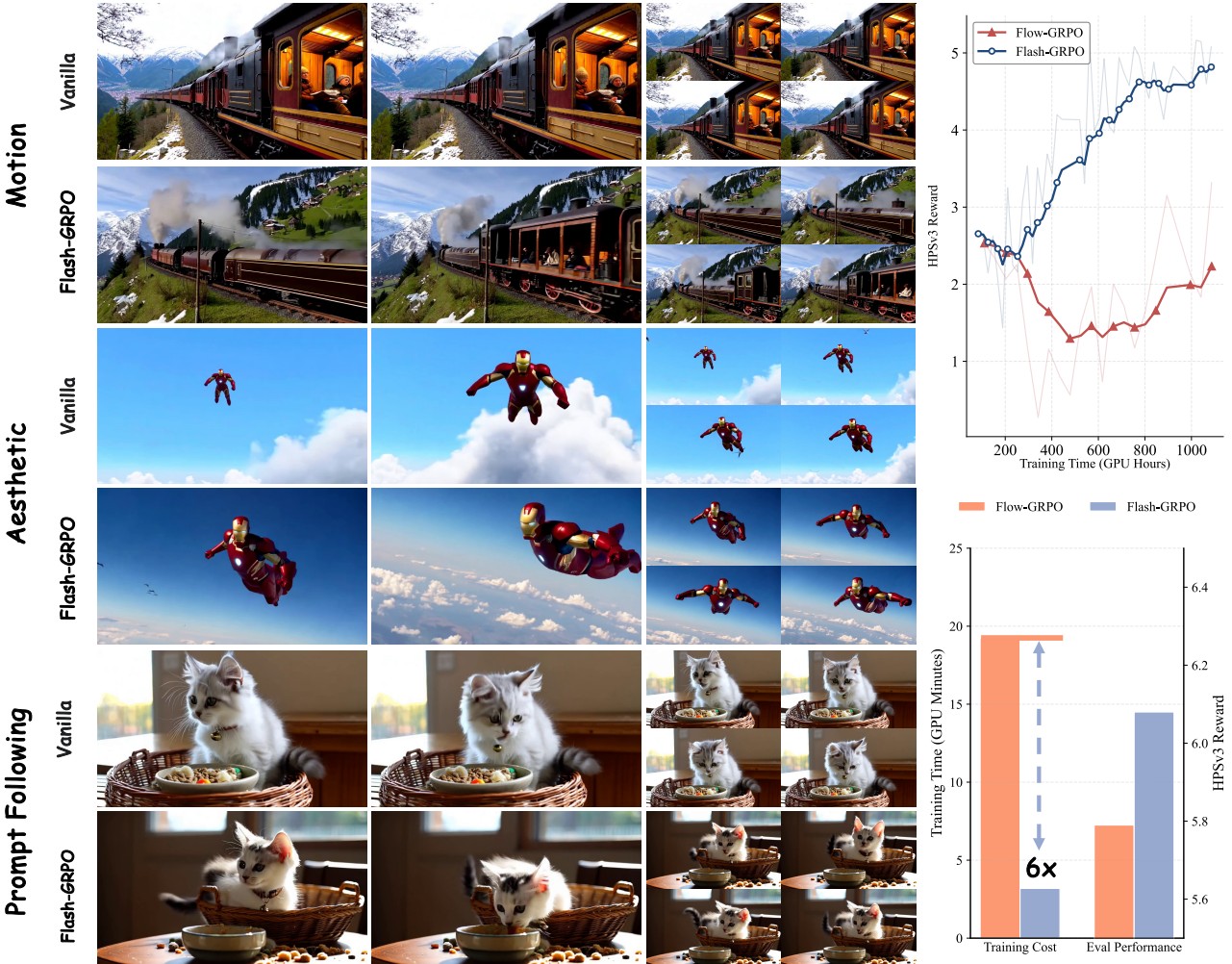

*Figure 1.* Overview of Flash-GRPO performance. **(Left)** Qualitative comparison across three dimensions: Motion, Aesthetic, and Prompt Following. Flash-GRPO generates videos with enhanced temporal dynamics (train sequence), improved visual quality (Iron Man), and better prompt adherence (cat with food bowl). **(Top Right)** Training reward curves showing that Flash-GRPO achieves stable monotonic improvement while Flow-GRPO exhibits slower convergence in training time. **(Bottom Right)** Efficiency comparison: Flash-GRPO achieves $6\times$ acceleration in training cost while attaining higher evaluation performance.

framework that achieves full trajectory performance using only one timestep per training. Our method addresses two fundamental challenges inherent to single-step optimization. The first challenge is timestep-confounded advantage estimation: a naive solution is to randomly assign timesteps within advantage groups, entangling reward variance with the intrinsic difficulty of different noise levels. To this end, we propose iso-temporal grouping, which enforces that all rollouts for a given prompt share the same timestep while varying only the initial noise. This factorizes the advantage computation, isolating policy-induced variance from timestep-induced variance and ensuring that relative performance comparisons occur under identical denoising conditions. Temporal diversity is preserved through stratified sampling across the global batch. The second challenge is gradient scale heterogeneity: we derive that the policy

gradient inherently contains a time-dependent scaling factor arising from the SDE discretization, which varies by orders of magnitude across the diffusion trajectory. This induces severe optimization imbalance where early timesteps dominate parameter updates regardless of their actual importance. We introduce temporal gradient rectification, which explicitly normalizes to unity, ensuring uniform contribution from all timesteps and eliminating discretization-induced bias from the optimization dynamics.

Together, these mechanisms enable Flash-GRPO to achieve single-step training with substantially reduced computational cost per iteration while maintaining training stability and reaching performance comparable to full-trajectory methods. Extensive experiments on both 1.3B and 14B video models validate that our approach eliminates the efficiency-quality trade-off, making high-quality video RL

alignment both practical and scalable. Our contributions are threefold:

- We identify two root causes of optimization instability in single-step video GRPO: timestep-confounded advantage estimation that entangles policy performance with noise level difficulty, and time-dependent gradient scaling that induces magnitude imbalance across the diffusion trajectory. We provide theoretical derivations and empirical validation for both phenomena.

- We propose Flash-GRPO, a principled single-step training framework that combines iso-temporal grouping for precise advantage estimation with temporal gradient rectification for balanced optimization, achieving full trajectory performance at minimal computational cost.

- We validate Flash-GRPO on video models from 1.3B to 14B parameters, demonstrating substantial training acceleration with consistent stability. Under equivalent computational budgets, Flash-GRPO outperforms both existing efficiency methods in stability and full trajectory training in alignment quality.

## 2. Related Work

**Video Diffusion Models.** Diffusion models have recently emerged as the dominant paradigm for video generation, capable of producing high-fidelity, temporally coherent sequences with superior controllability (Song et al., 2020; Dhariwal & Nichol, 2021; Song & Ermon, 2019). Early approaches, such as the Video Diffusion Model (VDM) (Ho et al., 2022), extended the 2D U-Net architecture to 3D to jointly model spatial and temporal dependencies. However, modeling directly in high-dimensional pixel space incurs prohibitive computational costs, which necessitated the development of latent space representations (Blattmann et al., 2023). More recently, the field has witnessed a significant architectural shift from standard U-Net designs (Rombach et al., 2022; Ho et al., 2022) to scalable Diffusion Transformers (DiT) (Peebles & Xie, 2023; Ma et al., 2024; Kong et al., 2024). Proprietary models such as Gen-3 (Runway, 2024) and Kling (Kuaishou, 2024) have set high benchmarks for visual fidelity and physical consistency. Concurrently, the open-source community has made substantial contributions, fostering powerful systems like CogVideoX (Yang et al., 2024), HunyuanVideo (Team, 2025) and Wan (Wan et al., 2025). While these models achieve impressive generation quality through large-scale pretraining, aligning them with human preferences via reinforcement learning has proven essential for further improving visual aesthetics, prompt adherence, and motion dynamics.

**Group Relative Policy Optimization.** Reinforcement learning has proven effective for aligning Large Language

Models with human preferences through methods such as PPO (Schulman et al., 2017) and DPO (Rafailov et al., 2023). Recent works have extended this paradigm to diffusion and flow-matching models for visual generation. Flow-GRPO (Liu et al., 2025a) and DanceGRPO (Xue et al., 2025) pioneered the application of GRPO to flow-matching by converting deterministic ODE sampling into stochastic SDE formulations for exploration. Several improvements have followed: MixGRPO (Li et al., 2025) accelerates training via hybrid ODE-SDE sampling; Flow-CPS (Wang & Yu, 2025) addresses noise coefficient inconsistencies to improve reward estimation; TempFlowGRPO (He et al., 2025) and $G^2$RPO (Guo et al., 2025) tackle credit assignment through temporal reward shaping. Despite these advances, existing methods predominantly focus on image generation, leaving the computational challenges of video alignment largely unexplored. Our work addresses this gap by proposing an efficient single-step training framework specifically designed for video diffusion models.

## 3. Preliminary

**Group Relative Policy Optimization for Flow Matching.** Flow-GRPO (Liu et al., 2025a) and DanceGRPO (Xue et al., 2025) pioneer the application of reinforcement learning to flow-matching models by adapting Group Relative Policy Optimization (GRPO) from the LLM domain. The core training objective maximizes the expected advantage over a group of rollouts:

$$\mathcal{J}_{\text{GRPO}}(\theta) = \mathbb{E}_{\boldsymbol{c} \sim \mathcal{C}, \{\boldsymbol{x}^i\}_{i=1}^{G} \sim \pi_{\theta_{\text{old}}}(\cdot|\boldsymbol{c})} \left[ f(r, \hat{A}, \theta, \varepsilon, \beta) \right], \tag{1}$$

where the objective function aggregates clipped policy ratios across all timesteps:

$$f(r, \hat{A}, \theta, \varepsilon, \beta) = \frac{1}{GT} \sum_{i=1}^{G} \sum_{t=0}^{T-1} \left( \min \left( r_t^i(\theta) \hat{A}_t^i, \right. \right.$$
$$\left. \left. \text{clip} \left( r_t^i(\theta), 1 - \varepsilon, 1 + \varepsilon \right) \hat{A}_t^i \right) - \beta D_{\text{KL}}(\pi_\theta \| \pi_{\text{ref}}) \right). \tag{2}$$

Here, $r_t^i(\theta) = \pi_\theta(\boldsymbol{x}_{t-1}^i|\boldsymbol{x}_t^i)/\pi_{\theta_{\text{old}}}(\boldsymbol{x}_{t-1}^i|\boldsymbol{x}_t^i)$ represents the policy ratio, $\hat{A}_t^i$ is the advantage estimate, and the summation over $T$ timesteps reflects the dense supervision paradigm—this full-trajectory requirement is precisely the computational bottleneck our method aims to eliminate.

**ODE-to-SDE.** A critical prerequisite for applying GRPO is the ability to sample diverse trajectories for robust advantage estimation. However, standard flow matching models employ a deterministic ordinary differential equation (ODE) for the forward process:

$$\boldsymbol{x}_{t+\Delta t} = \boldsymbol{x}_t + \boldsymbol{v}_\theta(\boldsymbol{x}_t, t)\Delta t, \tag{3}$$

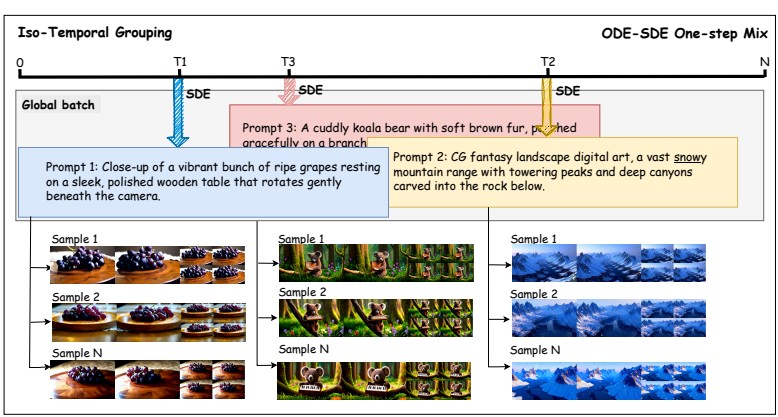 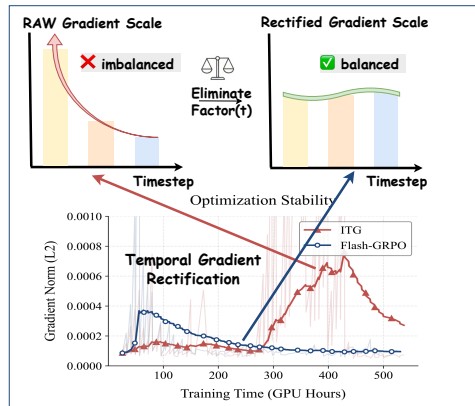

*Figure 2.* Overview of the Flash-GRPO Framework. (**Left**) Iso-temporal Grouping: each prompt performs ODE-to-SDE transition at a single sampled timestep for exploration and gradient computation, while other timesteps use deterministic ODE for accurate reward signals. Rollouts within each group share this transition timestep but differ in initial noise, factorizing policy-induced variance from timestep-induced variance. (**Right**) Temporal Gradient Rectification: the SDE discretization introduces a time-dependent scaling factor $\lambda(t)$ that causes gradient magnitudes to vary by orders of magnitude. Normalizing by $1/\lambda(t)$ ensures uniform contribution across timesteps, eliminating discretization-induced optimization bias.

which precludes the exploration necessary for RL. To enable stochastic rollouts while preserving the model's learned distribution, Flow-GRPO and DanceGRPO adopt an equivalent stochastic differential equation (SDE) formulation that matches the marginal probability $p_t(\boldsymbol{x})$ of the original ODE:

$$\boldsymbol{x}_{t+\Delta t} = \boldsymbol{x}_t +$$
$$\left[\boldsymbol{v}_\theta(\boldsymbol{x}_t, t) + \frac{\sigma_t^2}{2t}\left(\boldsymbol{x}_t + (1-t)\boldsymbol{v}_\theta(\boldsymbol{x}_t, t)\right)\right]\Delta t + \sigma_t\sqrt{\Delta t}\,\boldsymbol{\epsilon},$$
$$(4)$$

where $\boldsymbol{\epsilon} \sim \mathcal{N}(0, \boldsymbol{I})$ injects controlled stochasticity at noise level $\sigma_t$. This SDE framework provides the exploration mechanism required for GRPO while maintaining distributional equivalence to the pretrained model. Critically, this stochastic formulation introduces time-dependent scaling factors (embodied in the drift correction term $\frac{\sigma_t^2}{2t}$ and diffusion coefficient $\sigma_t$) that will later prove central to the gradient instability issues in one-step setting we address in Section 4.2.

## 4. Method

Our goal is to push training efficiency to its limit: optimizing only *one timestep* per rollout while matching full trajectory performance. Realizing this requires addressing two challenges that plague naive single-step approaches: (1) timestep-confounded variance in advantage estimation (Section 4.1), and (2) time-dependent gradient scale imbalances (Section 4.2).

### 4.1. Iso-Temporal Grouping for Precise Credit Assignment

Standard video generation pretraining achieves high efficiency by optimizing the vector field at a single randomly selected timestep per sample. To replicate this efficiency in the GRPO alignment phase, we adopt a single-step training paradigm. However, naively applying single-step GRPO to video models introduces a critical statistical challenge: *timestep-confounded reward variance*.

The fundamental issue lies in the inherent correlation between reward $R(\boldsymbol{x}_0, \boldsymbol{c})$ and noise level $t$. In a naive single-step strategy where each sample within a prompt group is assigned an independent random timestep, the group baseline becomes a mixture of rewards from varying noise levels:

$$\mu_{\text{naive}} = \frac{1}{G}\sum_{i=1}^{G} R(\boldsymbol{x}_0^i(\boldsymbol{x}_{t_i}), \boldsymbol{c}), \quad \text{where } t_i \sim \mathcal{U}[0, T] \quad (5)$$

This timestep heterogeneity acts as a confounding variable: the observed reward variance reflects both the policy's generation quality *and* the inherent difficulty of different timesteps. Consequently, advantage estimates become unstable and unreliable, undermining effective policy optimization.

To eliminate this confounding effect, we propose iso-temporal grouping. For a training batch of $B$ prompts $\{\boldsymbol{c}_k\}_{k=1}^B$, each prompt $\boldsymbol{c}_k$ is assigned a distinct timestep $t_k \sim \mathcal{U}[0, T]$. Within each prompt group, all $G$ rollouts share this same timestep $t_k$ but are initialized with different Gaussian noise $\boldsymbol{\epsilon}_i$:

$$\mathcal{G}_k = \{\boldsymbol{x}_{t_k}^i \mid i \in [1, G]\}, \tag{6}$$

Different prompt groups may have different timesteps, ensuring temporal diversity across the global batch. During denoising, each prompt group performs a single-step ODE-to-SDE transition at its assigned timestep $t_k$: the selected timestep uses SDE sampling (Equation 4) to enable exploration and gradient computation, while all other timesteps use deterministic ODE to produce higher-quality generations and more accurate reward signals. By enforcing identical timesteps within each prompt group, we *decouple policy performance from timestep difficulty*: samples within the same group are compared under identical denoising conditions, so the advantage reflects generation quality rather than timestep-dependent confounders.

For training, we compute the policy gradient only at the ODE-to-SDE transition timestep $t_k$ for each prompt group, ensuring that gradients incorporate diverse timesteps across the batch while maintaining precise advantage estimation within each group.

### 4.2. Temporal Gradient Rectification

While iso-temporal grouping stabilizes advantage estimation, a second critical challenge arises from the intrinsic structure of the policy gradient itself. We reveal that the gradient magnitude is implicitly modulated by time-dependent scaling factors, leading to severe optimization instability when training across diverse timesteps.

Critically, this imbalance is an artifact of the discretization scheme rather than a reflection of generation quality or reward signal strength. The uncalibrated variance in gradient scales is the theoretical root cause of the optimization instability observed in baseline methods. As illustrated in Figure 2, this manifests empirically as severe fluctuations in gradient norms, ultimately leading to catastrophic performance collapses in the reward curve.

To understand this phenomenon, we derive the explicit policy gradient for the reverse generation process. The standard reinforcement learning objective at timestep $t$ is:

$$\nabla_\theta \mathcal{J} = \mathbb{E}_{\boldsymbol{x}_t, \boldsymbol{\epsilon}} \left[ \hat{A}_t \cdot \nabla_\theta \log p_\theta(\boldsymbol{x}_{t-1}|\boldsymbol{x}_t) \right]. \quad (7)$$

Under the Gaussian transition kernel induced by the Euler-Maruyama discretization of the reverse-time SDE, the previous state $\boldsymbol{x}_{t-1}$ is modeled as:

$$\boldsymbol{x}_{t-1} = \underbrace{\boldsymbol{\mu}_\theta(\boldsymbol{x}_t, t)}_{\text{Mean}} + \underbrace{\sigma_t \sqrt{\Delta t}}_{\text{Std}} \cdot \boldsymbol{\epsilon}, \quad (8)$$

where the predicted mean $\boldsymbol{\mu}_\theta$ is parameterized by the learned vector field $\boldsymbol{v}_\theta$:

$$\boldsymbol{\mu}_\theta(\boldsymbol{x}_t, t) =$$
$$\boldsymbol{x}_t + \left[ \boldsymbol{v}_\theta(\boldsymbol{x}_t, t) + \frac{\sigma_t^2}{2t} \left( \boldsymbol{x}_t + (1-t)\boldsymbol{v}_\theta(\boldsymbol{x}_t, t) \right) \right] \Delta t. \quad (9)$$

Substituting this into the score function and expanding the gradient term yields:

$$\nabla_\theta \log p_\theta(\boldsymbol{x}_{t-1}|\boldsymbol{x}_t)$$
$$= \nabla_\theta \left( -\frac{\|\boldsymbol{x}_{t-1} - \boldsymbol{\mu}_\theta(\boldsymbol{x}_t, t)\|^2}{2\sigma_t^2 \Delta t} \right)$$
$$= \frac{\boldsymbol{x}_{t-1} - \boldsymbol{\mu}_\theta(\boldsymbol{x}_t, t)}{\sigma_t^2 \Delta t} \nabla_\theta \boldsymbol{\mu}_\theta(\boldsymbol{x}_t, t)$$
$$= \frac{\sigma_t \sqrt{\Delta t} \boldsymbol{\epsilon}}{\sigma_t^2 \Delta t} \nabla_\theta \boldsymbol{\mu}_\theta(\boldsymbol{x}_t, t)$$
$$= \frac{\boldsymbol{\epsilon}}{\sigma_t \sqrt{\Delta t}} \cdot \Delta t \left( 1 + \frac{\sigma_t^2(1-t)}{2t} \right) \nabla_\theta \boldsymbol{v}_\theta(\boldsymbol{x}_t, t)$$
$$= \underbrace{\left( \frac{\sqrt{\Delta t}}{\sigma_t} + \frac{\sigma_t \sqrt{\Delta t}(1-t)}{2t} \right)}_{\lambda(t): \text{ Time-dependent Scaling}} \boldsymbol{\epsilon} \cdot \nabla_\theta \boldsymbol{v}_\theta(\boldsymbol{x}_t, t). \quad (10)$$

Equation 10 reveals a critical structural issue: the policy gradient is intrinsically scaled by a time-dependent coefficient $\lambda(t) = \frac{\sqrt{\Delta t}}{\sigma_t} + \frac{\sigma_t \sqrt{\Delta t}(1-t)}{2t}$. **In our Flash-GRPO framework, where different prompts within a batch are trained at distinct timesteps, $\lambda(t)$ acts as an implicit, heterogeneous weighting factor.** As $\sigma_t$ and $t$ vary across the diffusion trajectory, $\lambda(t)$ can fluctuate by orders of magnitude—prompts sampled at different timesteps thus contribute to the parameter update with vastly inconsistent magnitudes.

To resolve this pathology, we propose **Temporal Gradient Rectification**, which explicitly normalizes the time-dependent scaling factor. Specifically, we rescale the gradient by $1/\lambda(t)$, effectively setting $\lambda(t) \to 1$ for all timesteps. The **uncliped** rectified policy loss is:

$$\mathcal{L}_{\text{TGR}}(\theta) = \frac{1}{G} \sum_{i=1}^{G} \frac{\hat{A}_t^i}{\lambda(t)} \cdot r_t^i(\theta), \quad (11)$$

where $\lambda(t) = \frac{\sqrt{\Delta t}}{\sigma_t} + \frac{\sigma_t \sqrt{\Delta t}(1-t)}{2t}$ is the time-dependent scaling factor derived in Equation 10. By decoupling the optimization dynamics from the sampler's discretization scale, this rectification ensures that all prompts contribute equally to the parameter update, regardless of their position in the diffusion trajectory. The result is dramatically enhanced training stability and consistent monotonic reward growth, as validated in our experiments.

## 5. Experiment

### 5.1. Experimental Setup

**Datasets and Models.** Following the setting in Dance-GRPO (Xue et al., 2025), we utilize their prompt dataset for

*Table 1.* Detailed comparison of **General Video Quality** using VBench metrics. We evaluate aesthetic quality, image quality, subject consistency, and object class to ensure the RL fine-tuning retains the generative capability of the backbone model. We reproduce the official VBench results; * indicates our own reproduction results (mismatch). Best scores are in blue .

| Method | GPU Hours | Aesthetic Quality ↑ | Imaging Quality ↑ | Subject Consistency ↑ | Object Class ↑ |
|---|---|---|---|---|---|
| CogVideoX-2B (Yang et al., 2024) | – | 61.07 | 62.37 | 96.52 | 86.48 |
| Hunyuan-Video (Kong et al., 2024) | – | 60.36 | 67.56 | 97.37 | 86.10 |
| Wan2.1-T2V-1.3B (Wan et al., 2025) | – | 65.46 | 66.79*/67.01 | 97.56 | 88.84*/88.81 |
| **Flow-GRPO-Fast1** | 350 | 65.92 | 65.96 | 98.46 | 88.15 |
| **Flow-GRPO** | 350 | 65.79 | 68.60 | 97.28 | 87.92 |
| **Flash-GRPO** | 350 | 66.43 | 68.28 | 98.70 | 90.00 |

training, while holding out a distinct split of 300 prompts for evaluation. We employ the Wan2.1 family (Wan et al., 2025) as our foundation models, validating our method on both the 1.3B and the large-scale 14B variants.

**Implementation Details.** We tailor the sampling schedule during training: we utilize 20 sampling steps for the 1.3B model and an accelerated 12 sampling steps for the 14B model. The classifier-free guidance (CFG) scale is fixed at 4.5. To ensure stable policy updates under the single-step training paradigm, we enforce a strict GRPO clip ratio of 0.001. Meanwhile, we benchmark our method against two established baselines: Flow-GRPO and Flow-GRPO-Fast. **Baselines.** For Flow-GRPO, we adopt the official video RL configuration, which restricts training to the first half of denoising timesteps. For efficiency methods, **it is worth noting** that Flow-GRPO-Fast's few-step training mechanism is conceptually aligned with MixGRPO. We therefore evaluate Flow-GRPO-Fast under a single-step update setting, denoted as Flow-GRPO-Fast1, to directly compare with our single-step framework.

**Evaluation.** For the held-out evaluation set, we perform inference using 50 sampling steps to assess the model's generation capability. We evaluate the generated videos across two primary dimensions: Visual Quality and Motion Quality. **Visual Quality.** We adopt HPSv3 (Ma et al., 2025) as the reward model for visual quality assessment. Following (Team et al., 2025), we calculate reward scores for all sampled frames and compute the advantage based on the average of the top 30% scoring frames, which mitigates the impact of low rewards caused by content inconsistency during temporal transitions. **Motion Quality.** We employ the motion score from VideoAlign (Liu et al., 2025b) to evaluate temporal coherence and motion dynamics. This metric specifically captures the smoothness and physical plausibility of generated motion sequences. **General Video Quality.** We further evaluate on VBench (Huang et al., 2024) to assess overall video quality across multiple dimensions including aesthetic appeal, imaging fidelity, and

semantic consistency. **Additional quantitative analysis and experiments are provided in Appendix A and B.**

### 5.2. Performance on VBench Quality Metrics

We evaluated the performance of our method on the VBench benchmark (Huang et al., 2024). Adhering to the official VBench evaluation protocol, we utilized both enhanced prompts and negative prompts, while ensuring all other parameters remained consistent with the standard VBench settings. Table 1 summarizes performance on VBench metrics, which assess video quality across aesthetic appeal, imaging fidelity, and semantic consistency. With 350 GPU hours of training on Wan2.1-T2V-1.3B, Flash-GRPO achieves the highest Aesthetic Quality (66.43) and Subject Consistency (98.70), outperforming both Flow-GRPO-Fast1 and Flow-GRPO. Notably, Flow-GRPO-Fast1 suffers degraded Imaging Quality (65.96) compared to full trajectory Flow-GRPO (68.60), reflecting the cost of naive subsampling. Flash-GRPO maintains strong Imaging Quality (68.28) while achieving superior efficiency, demonstrating that our method decouples computational cost from alignment quality. Compared to CogVideoX-2B and Hunyuan-Video, all methods based on Wan2.1 achieve substantial improvements in Aesthetic Quality, with Flash-GRPO reaching the highest score. All methods maintain high consistency metrics ($\geq 97$), confirming that RL fine-tuning preserves the backbone's generative capabilities.

### 5.3. Visual Comparison.

Figure 3 presents visual comparisons between the vanilla Wan2.1 baseline and Flash-GRPO. We observe consistent improvements across diverse scenes and styles. In the savanna scene (rows 1-2), the baseline produces flickering artifacts in the grass region (red box), while Flash-GRPO maintains stable background throughout the sequence. For the animated panda scene (rows 3-4), Flash-GRPO generates smoother character movements and more consistent facial expressions. In the cartoon animal scene (rows 5-

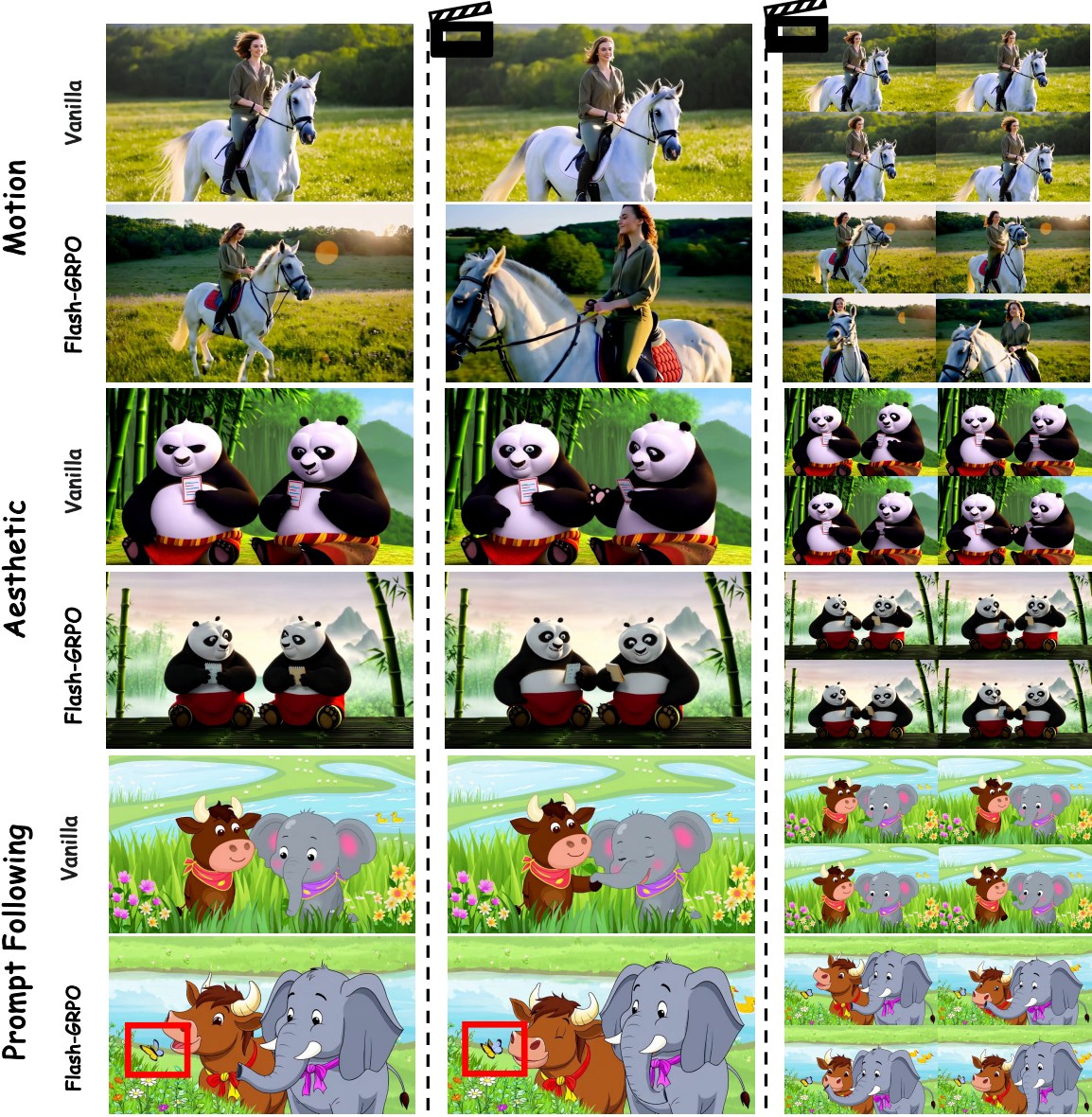

*Figure 3.* Qualitative comparison between vanilla Wan2.1 (odd rows) and Flash-GRPO (even rows) across three dimensions: Motion, Aesthetic, and Prompt Following. Flash-GRPO produces videos with enhanced temporal dynamics (horse riding sequence), improved visual quality and richer details (panda scene), and better prompt adherence with additional elements (cartoon animals with butterfly, highlighted in red boxes).

6), the baseline exhibits unstable elements marked by the red box, whereas Flash-GRPO preserves spatial coherence across frames. These results demonstrate that Flash-GRPO effectively improves both visual quality and temporal consistency without sacrificing the generative diversity of the backbone model.

### 5.4. Ablation Study

We conduct ablation experiments to validate the contribution of each component in Flash-GRPO, starting from naive single-step training as baseline and incrementally adding iso-temporal grouping and temporal gradient rectification. As shown in Table 2, ***iso-temporal grouping*** alone provides notable improvement over the naive baseline by enforcing that all rollouts within a prompt group share the same timestep, disentangling advantage estimates from timestep difficulty and reducing variance in credit assignment. ***Temporal gradient rectification*** yields further gains, particularly in optimization stability: without rectification, gradient norms exhibit severe fluctuations due to the time-dependent scaling factor $\lambda(t)$, while normalizing $\lambda(t)$ eliminates these

*Table 2.* Ablation study on Wan2.1-1.3B with HPSv3 reward. ITG: Iso-temporal Grouping. TGR: Temporal Gradient Rectification.

| Method | Train Stability | Eval Reward |
|---|---|---|
| Wan2.1-1.3B | - | 4.67 |
| Naive Single-step | ✗ | 4.64 |
| + ITG | ✗ | 5.31 |
| + ITG + TGR (Full) | ✓ | 5.42 |

spikes and produces consistent gradient magnitudes across all timesteps.

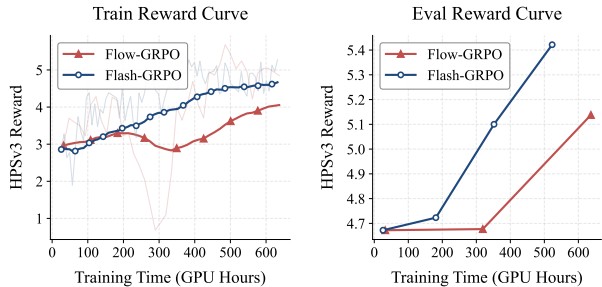

*Figure 5.* Comparison with full trajectory Flow-GRPO on HPSv3. Flash-GRPO achieves faster convergence and higher reward ceiling on both training (Left) and evaluation (Right).

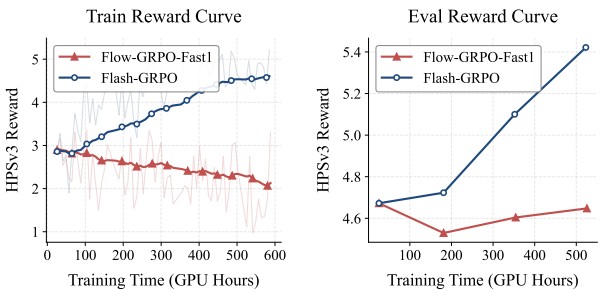

*Figure 4.* HPSv3 reward curves. Flow-GRPO-Fast1 suffers from optimization collapse on both training (Left) and evaluation (Right), while Flash-GRPO maintains stable convergence.

### 5.5. Analysis

**Comparison with Flow-GRPO-Fast1.** Flow-GRPO-Fast and MixGRPO adopt a sliding window approach to reduce computational overhead. We evaluate Flow-GRPO-Fast with window size 1 (denoted Fast1) under two training regimes. ***Without KL regularization*** (Figure 4), Fast1 exhibits catastrophic failure with severe variance and persistent decline, while Flash-GRPO achieves robust monotonic reward growth, validating that temporal gradient rectification alone suffices to stabilize single-step training.

**Comparison with Flow-GRPO.** We benchmark Flash-GRPO against full trajectory Flow-GRPO. Due to prohibitive computational costs, we limit this comparison to the first half of the training schedule. As shown in Figure 5, Flow-GRPO suffers persistent instability with high variance and catastrophic collapse between 200-400 GPU hours, while Flash-GRPO maintains stable monotonic improvement throughout. On the evaluation curve (Right), Flash-GRPO demonstrates steeper ascent and reaches higher quality earlier, achieving peak reward of approximately 5.4 (versus 5.1 for Flow-GRPO). These results suggest that our single-step framework is a more robust alternative for video alignment under low computational budgets.

**Scalability to 14B Models.** We validate Flash-GRPO on the 14B parameter Wan2.1 model, where the optimization

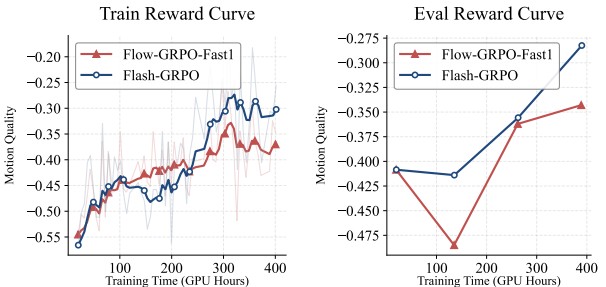

*Figure 6.* Motion Quality evaluation. Flash-GRPO achieves stable improvement and higher final performance on both training (Left) and evaluation (Right) sets, indicating superior learning of temporal coherence compared to Flow-GRPO-Fast1.

landscape becomes more slower for human preference alignment. As shown in Figure 1, Flash-GRPO maintains consistent stability and monotonic growth at this scale, while Flow-GRPO exhibits slower growth as the expanded parameter space amplifies the cost of training. This demonstrates that Flash-GRPO becomes an effective way to obtain higher alignment under low computational budgets.

**Motion Quality.** We further evaluate Motion Quality to assess temporal coherence and dynamic consistency. Figure 6 shows that Flow-GRPO-Fast1 exhibits similar instability patterns on motion metrics. Flash-GRPO maintains stable improvement, achieving a final score of approximately $-0.28$ compared to $-0.34$ for the baseline. This confirms that Flash-GRPO improves both visual aesthetics and temporal dynamics.

## 6. Conclusion

We presented Flash-GRPO, a framework that enables single-step training to match full-trajectory performance for video RL alignment. Our investigation identifies two primary sources of instability in single-step video RL: first, mixing timesteps within advantage groups confounds reward variance with timestep difficulty, obscuring true policy performance; second, the inherent time-dependent scaling factor in policy gradients causes vastly inconsistent update magni-

tudes across timesteps. Flash-GRPO resolves both through iso-temporal grouping and gradient rectification, achieving stable optimization without computational overhead. Experiments across 1.3B to 14B models validate the effectiveness and scalability of this approach, substantially reducing training costs while preserving alignment quality comparable to full-trajectory methods.

## Impact Statement

This paper tackles the computational bottleneck in diffusion RL by significantly accelerating training efficiency. This advancement facilitates faster policy iteration in high-dimensional tasks and promotes environmental sustainability by reducing the energy footprint of intensive RL experiments. Furthermore, by lowering hardware barriers, our method supports the democratization of AI research, enabling broader participation in developing state-of-the-art generative policies for robotics and autonomous systems.

The potential positive impacts include more resource-efficient development of complex agents in fields like robotics and industrial automation. The ethical implications are consistent with general generative modeling and RL, as our approach improves efficiency without introducing novel risks beyond existing challenges in algorithmic bias or AI safety. No further broader impacts are identified.

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

## A. More Experiments Comparison with Flow-GRPO-Fast1.

***With KL regularization*** (Figure 7), KL loss prevents Fast1 from collapsing but a substantial performance gap persists: Flash-GRPO converges faster, reaches a higher ceiling, and achieves approximately 5.35 on HPSv3 versus Fast1's 4.9 on the held-out set.

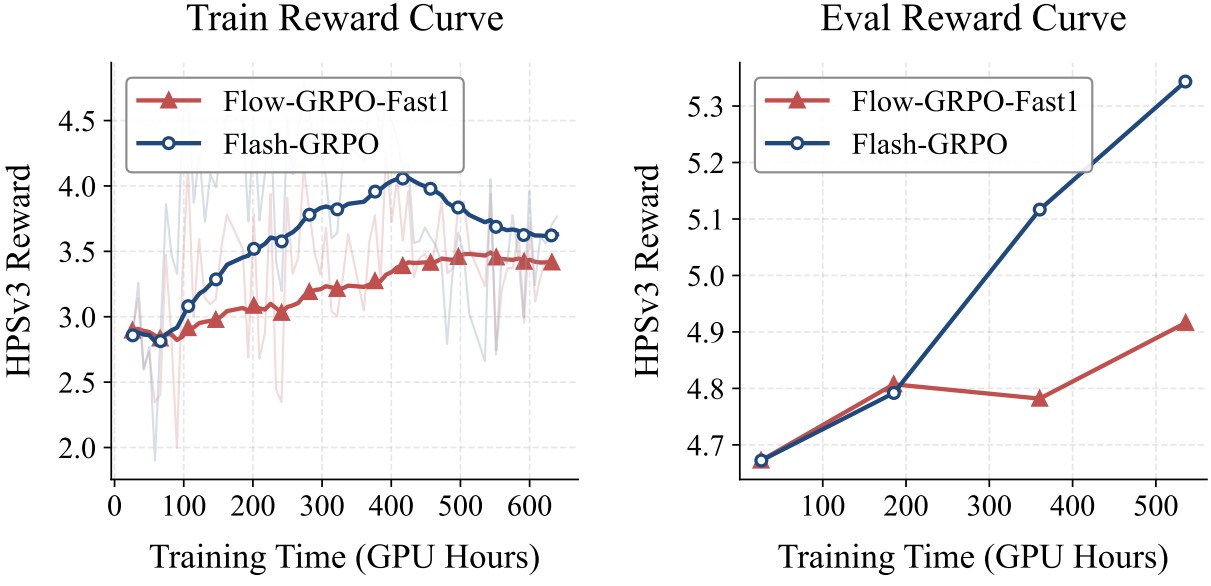

*Figure 7.* HPSv3 reward curves with KL regularization. Flash-GRPO achieves faster convergence and higher performance ceiling on both training (Left) and evaluation (Right), while Flow-GRPO-Fast1 plateaus early with limited generalization.

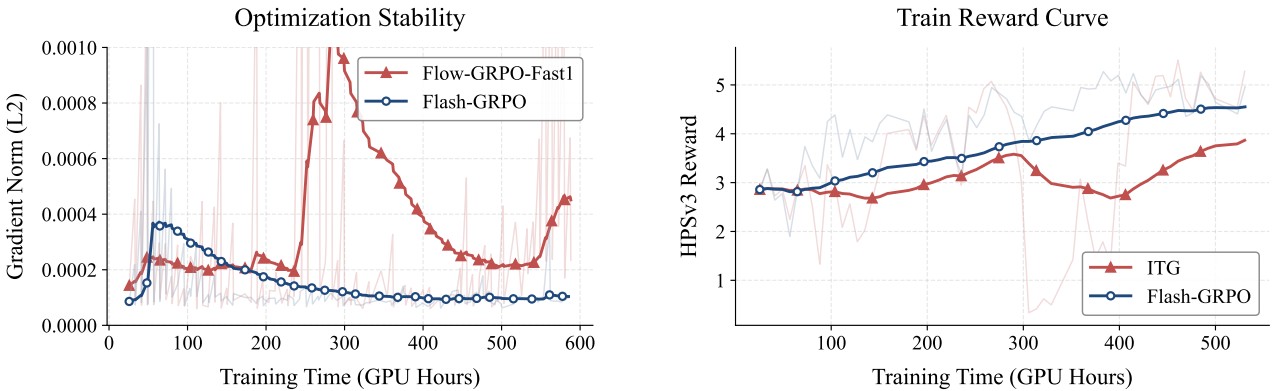

*Figure 8.* Analysis of Training Stability and Convergence. **(Left)** Optimization Stability Analysis without KL Regularization. We visualize the evolution of the gradient norm during training. **Red (Flow-GRPO-Fast1)**: Without KL constraints, the baseline suffers from severe optimization instability, evidenced by catastrophic gradient spikes and high variance. **Blue (Ours)**: In contrast, our method maintains a consistently low and stable gradient norm, demonstrating that our gradient rectification strategy effectively regularizes the optimization landscape even in the absence of explicit KL penalties. **(Right)** Reward Curve: The instability in the baseline leads to a catastrophic performance drop (reward collapse) around 300 GPU hours. Flash-GRPO ensures monotonic reward growth and achieves a significantly higher convergence ceiling.

***More Results.*** The left of Figure 8, We further visualize the gradient norm trajectories during training in Figure 4. In the unconstrained setting without KL regularization, the Flow-GRPO-Fast method exhibits severe optimization instability, evidenced by catastrophic gradient spikes and high variance. Conversely, Flash-GRPO maintains a consistently low and stable gradient norm throughout the process. This result indicates that even in the absence of explicit KL penalties, our temporal gradient rectification strategy effectively regularizes the optimization landscape.

## B. Impact of Temporal Gradient Rectification.

The right of Figure 8 compares training reward curves with and without our rectification strategy. Applying temporal gradient rectification leads to a significantly more stable trajectory. In contrast, the unrectified baseline suffers from severe optimization instability, evidenced by a catastrophic reward collapse between 300 and 400 GPU hours.

## C. Algorithm of Flash-GRPO.

---

**Algorithm 1** Flash-GRPO (Take a prompt $c$ as case)

---

**Require:** Prompt $c$, group size $G$, total timesteps $T$, reward models $R$.
**Ensure:** Optimized policy parameters $\theta$
1: Initialize policy parameters $\theta$, reference policy $\pi_{\text{ref}}$
2: **repeat**
3:    // Sample
4:    Random sample a timestep $k$ for prompt $c$
5:    **for** $t = T$ **to** $0$ **do**
6:       **if** $t == k$ **then**
7:          $\boldsymbol{x}_{t-1} = \boldsymbol{x}_t + [\boldsymbol{v}_\theta(\boldsymbol{x}_t, t) + \frac{\sigma_t^2}{2t}(\boldsymbol{x}_t + (1-t)\boldsymbol{v}_\theta(\boldsymbol{x}_t, t))]\Delta t + \sigma_t\sqrt{\Delta t}\boldsymbol{\epsilon}$ // Equation 4
8:       **else**
9:          $\boldsymbol{x}_{t-1} = \boldsymbol{x}_t - \boldsymbol{v}_t dt$ // Equation 3
10:      **end if**
11:   **end for**
12:   // Compute Advantages
13:   Compute mean, std of $\{R(\boldsymbol{x}_0^i, \boldsymbol{c})\}_{i=1}^G$ and $\{A(\boldsymbol{x}_0^i, \boldsymbol{c})\}_{i=1}^G$
14:   // Training
15:   $\mathcal{L}_{\text{total}} = \mathcal{L}_{\text{TGR}}(\theta)$ // Equation 11
16:   $\theta \leftarrow \theta - \eta\nabla_\theta\mathcal{L}_{\text{total}}$
17: **until** convergence

---

## D. More Qualitative Evaluation

We present qualitative comparisons between Flash-GRPO and vanilla Wan2.1 on both 1.3B and 14B models. As shown in Figures 9 - 12, Flash-GRPO consistently generates videos with higher visual fidelity, richer scene details, and smoother motion dynamics.

On the 1.3B model (Figure 9), the waterfall scene demonstrates that Flash-GRPO produces more coherent human motion in the foreground region (red boxes). For character animations, Flash-GRPO achieves more realistic rendering with improved lighting and texture details: in the cooking scene, facial features, kitchen environment, and the watermelon cutting action are noticeably enhanced.

On the 14B model (Figures 10 -11), Flash-GRPO shows consistent improvements across diverse scenes. The Japanese garden scene exhibits more stable prompt following with the foreground object and enhanced depth-of-field effects. The bird and sailboat sequences display more fluid motion. Animal scenes maintain correct semantic representation with richer environmental details (red boxes). In Figure 12, the cat sequence shows improved motion aesthetics, while the dog chasing scene and the hand-held sword CG scene demonstrate Flash-GRPO's superior prompt following capability.

These results confirm that Flash-GRPO effectively improves visual aesthetics, temporal coherence, and prompt adherence across different model scales and content types.

We present comprehensive qualitative comparisons to demonstrate the superior quality achieved by our method. The visualization results consistently show that our approach generates videos with enhanced fidelity, better motion smoothness, and prompt following to complex prompts, and fewer visual artifacts compared to vanilla.

**Prompts in Figure 1.** The prompts in Figure 1 are as follows:

1. A vintage steam train slowly moving along a winding mountain track. The train is painted in faded red and black colors, with steam billowing out from its chimney. The landscape is covered in snow-capped peaks and lush greenery. Trees sway gently in the wind, their branches touching the sides of the train. The carriage interiors are dimly lit, with wooden panels and brass fittings. Passengers inside, bundled up in woolen coats and hats, sit quietly, some reading newspapers, others sleeping. The camera captures the train as it steadily climbs the mountain, capturing the steam rising into the crisp mountain air. The background features a serene, snowy mountain range with a few distant villages nestled at the base. Low-angle shot, medium shot of the train partially visible.

2. Marvel superhero Iron Man flying high in the sky, amidst a clear blue cloudless day. Tony Stark, wearing his iconic red, gold, and black armor, pilots the Iron Man suit effortlessly. His sleek helmet reflects the sunlight, and his glowing red eyes scan the horizon. Flying at an altitude of over 10,000 feet, he performs acrobatic maneuvers, twisting and turning gracefully. The Iron Man suit's thrusters emit a soft humming sound as it glides smoothly. The background showcases vast, unobstructed skies dotted with fluffy white clouds. In the distance, a few birds fly by, adding life to the serene landscape. Iron Man maintains a calm and focused expression, ready for any challenge. The shot captures him from above, showcasing the intricate design and movement of his suit. Dynamic aerial perspective, fast-paced camera movements, and sweeping shots reveal the beauty and power of Iron Man's flight.

3. A gentle scene captured in soft focus, a fluffy white kitten with oversized green eyes and tufted ears sits contentedly on a woven basket. The kitten's fur is a mix of soft gray and creamy white, with occasional specks of black. It wears a small, cozy brown collar adorned with a tiny bell. The kitten is surrounded by a variety of colorful cat treats scattered in a ceramic bowl on a wooden table. The bowl is filled with wet food, partially consumed, with bits of kibble still visible. The kitten's tail curls gently as it eats, occasionally batting at stray crumbs with its paw. The background is a softly lit room, with soft shadows highlighting the textures of the furniture and floor. A window behind the scene shows a sunny afternoon outside. The scene is captured with a warm, nostalgic feel, reminiscent of old family photos. Soft focus, medium shot, half-body view.

**Prompts in Figure 3.**  The prompts in Figure 3 are as follows:

1. A person riding a majestic white horse, their attire blending seamlessly with the horse's coat, both animals moving gracefully across a rolling green pasture. The person has tousled brown hair and expressive blue eyes, their posture confident and relaxed as they hold the reins with steady hands. The horse's mane flows freely, catching the morning sunlight, adding a vibrant glow to the scene. In the background, lush trees and wildflowers dot the landscape, creating a serene and picturesque atmosphere. The person is wearing a simple yet elegant olive green tunic and sturdy leather boots, perfectly suited for the outdoors. The camera follows the duo, capturing moments of interaction between the rider and the horse, including subtle nods and playful glances. The scene transitions from a wide shot of the horse and rider to a medium shot focusing on the person's face, highlighting their joy and connection with nature. Cinematic lighting with soft shadows and warm tones enhances the emotional depth of the moment.

2. CG animation digital art, two adorable pandas sitting side-by-side on a bamboo forest backdrop. The pandas have expressive faces, one looking thoughtful with a raised eyebrow, the other with a curious look. They are both wearing traditional panda costumes with bright red sashes tied around their waists. Each panda holds a small notebook in front of them, depicting an academic paper. The background features lush bamboo forests and misty mountain peaks. The pandas are engaged in animated conversation, occasionally pointing at their notes. Soft lighting casts a warm glow over the scene. Detailed digital artwork with realistic textures. Low-angle view, medium shot side-by-side seating.

3. A whimsical animated short film, featuring a gentle brown cow and a majestic gray elephant standing together in a lush green meadow. The cow has soft, curly horns and a friendly expression, while the elephant has wrinkled skin and large, wise eyes. They are both wearing colorful cloths tied around their necks. The meadow is filled with blooming wildflowers and butterflies fluttering around them. The cow is grazing on some grass nearby, while the elephant gently blows leaves off a tree branch. In the background, a river can be seen flowing peacefully, with ducks swimming gracefully. The scene is captured in vibrant pastel colors with soft lighting, emphasizing the close bond between the two animals. The animation style is hand-drawn with smooth, fluid lines. Medium shot, side-by-side composition.

**Prompts in Figure 9.**  The prompts in Figure 9 are as follows:

1. A tranquil tableau of a rugged cliff standing tall against a backdrop of a vast, clear blue sky dotted with fluffy white clouds. The cliff face is weathered and rocky, with moss and wildflowers clinging to its crags. A gentle breeze rustles the leaves of ancient pine trees that hug the cliff's edge. In the foreground, a small waterfall cascades down, creating a serene stream that meanders between smooth, rounded boulders. The scene is bathed in warm golden sunlight, casting long shadows and highlighting the intricate textures of the cliff. The atmosphere is calm and peaceful, with a hint of mystery. A lone hiker, dressed in muted colors, pauses at the base of the cliff, gazing upwards with a sense of awe and wonder. The hiker stands with one hand resting on a large rock, capturing the tranquility and beauty of the moment. Soft natural sounds of birds chirping and leaves rustling fill the air. High angle shot focusing on the entire cliff, then medium shot focusing on the hiker.

2. CG game concept digital art, a person in a casual outfit, cutting a large ripe watermelon with a clean and precise knife. The person has short messy brown hair and expressive eyes, wearing a white tank top and jeans. They are standing in a well-lit kitchen, surrounded by various fruits and vegetables. The watermelon is a vibrant shade of green with a few small black seeds visible. The person's hands are steady as they carefully cut the melon, revealing juicy slices. The background features modern kitchen appliances and colorful fruit arrangements. The lighting highlights the textures of the watermelon and the person's hands. Low-angle close-up shot, medium shot of the person and the watermelon.

**Prompts in Figure 10.** The prompts in Figure 10 are as follows:

1. A close-up shot of a traditional Japanese bamboo-handled wooden spoon, delicately crafted with intricate patterns etched into the wood. The spoon rests on a small wooden stand with a smooth, polished surface. The background is a blurred image of a serene Japanese garden, featuring lush greenery, cherry blossom trees, and a gentle stream. Soft lighting highlights the textures and craftsmanship of the spoon. The scene exudes a sense of tranquility and simplicity. Smooth, hand-drawn cel-shaded animation style. Close-up, low-angle view.

**Prompts in Figure 11.** The prompts in Figure 11 are as follows:

1. CG animation digital art, a majestic bird soaring gracefully in the clear blue sky. The bird has iridescent feathers with hints of purple and green, large wings spread wide, and sharp talons. It soars effortlessly with a serene expression, its gaze fixed towards the horizon. The background features fluffy clouds drifting lazily across the vast sky, with gentle sunlight casting a warm glow. The scene is captured from a high-angle perspective, showcasing the bird's magnificent flight path. Dynamic camera movement follows the bird's ascent, capturing its fluid motion and breathtaking view. Smooth lines and vibrant colors enhance the ethereal atmosphere.

2. CG game concept digital art, a serene boat sailing smoothly on a tranquil lake. The boat is a wooden sailboat with a white hull and black sails, reflecting the sunlight gently. The lake is a deep blue, with gentle ripples caused by the boat's passage. Trees along the shore sway gently in the breeze. The sky is a soft pastel shade of blue, with fluffy white clouds. The sun sets behind the trees, casting a warm orange glow over the scene. The boat is mid-lake, centered, with the captain standing at the helm, steering confidently. He wears a navy blue life jacket, a straw hat, and casual trousers. His face shows determination and joy as he guides the boat. The background features lush greenery and a peaceful atmosphere. Low-angle view, focusing on the captain and the boat.

3. A vibrant African wildlife scene captured in a documentary style, featuring a majestic zebra and a graceful giraffe standing side-by-side in a lush green savanna. The zebra has distinctive black and white stripes, while the giraffe boasts a long neck and spotted coat. Both animals are perched on soft grasses, with their eyes fixed on something in the distance. The zebra stands confidently, alert and curious, while the giraffe grazes calmly. The savanna landscape is filled with various flora and fauna, including colorful flowers, small birds, and a few distant elephants. The sun sets behind them, casting a warm golden hue over the scene. The composition includes a mix of wide and tight shots, showcasing the interaction between the two magnificent creatures. Documentary-style cinematography with natural lighting and subtle camera movements. Medium shot of the zebra and giraffe together, followed by wide shot of the savanna backdrop.

**Prompts in Figure 12.** The prompts in Figure 12 are as follows:

1. A playful feline sprinting joyfully across a lush green meadow dotted with wildflowers. The cat has sleek fur, expressive green eyes, and a fluffy tail that wags excitedly as it bounds forward. The meadow stretches out behind it, with vibrant sunflowers and buttercups swaying gently in the breeze. The sky above is a bright azure, filled with fluffy white clouds. The cat's joyful run is captured from a dynamic low-angle perspective, showcasing its agility and boundless energy. The scene is bathed in warm golden light, enhancing the cat's lively demeanor. Grass and petals trail behind the cat as it dashes towards the horizon. The background features a serene rural landscape, with small cottages and winding country roads visible in the distance. The overall composition is energetic and full of life, perfectly capturing the essence of a cat running happily.

2. A playful golden retriever fetching a ball in a lush green field. The dog has a shiny coat, expressive brown eyes, and floppy ears. It is wagging its tail excitedly as it chases after the bouncing ball. The field is dotted with wildflowers, creating a vibrant tapestry of colors. The sun shines brightly in the clear blue sky, casting a warm glow over everything. In the background, a small farmhouse can be seen nestled among the trees. The scene is captured with a dynamic camera movement, alternating between wide shots of the dog bounding across the field and close-ups of the dog's joyful expressions. Soft natural lighting enhances the mood, making the image feel alive and inviting. High-resolution, cinematic quality video. Wide shot establishing view followed by medium close-up shots of the dog's joyful moments.

3. CG game concept digital art, a sharp blade made of obsidian, glowing with an eerie blue light. The knife is held in one hand, the fingers wrapped tightly around the handle. The blade is curved, with intricate patterns etched along the edge. The knife glows softly, casting shadows and highlights. The handle is adorned with small crystals, each emitting a faint, pulsating light. The blade is coated in a thin layer of oil, giving it a slick and metallic sheen. The background is a dimly lit underground cavern, with stalactites hanging from the ceiling and flickering torches casting dancing shadows. The knife is held at a low angle, emphasizing the sharpness and weight. Close-up, low-angle view.

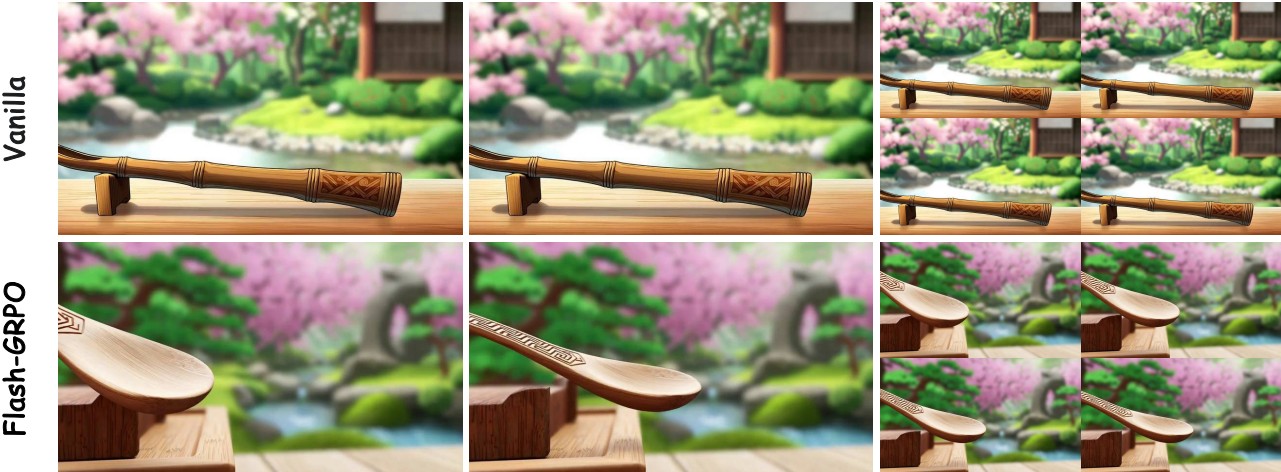

*Figure 9.* Qualitative comparison between Flash-GRPO and Vanilla with HPSv3 rewards on VBench prompts (Wan1.3B).

*Figure 10.* Qualitative comparison between Flash-GRPO and Vanilla with HPSv3 rewards on VBench prompts (Wan14B).

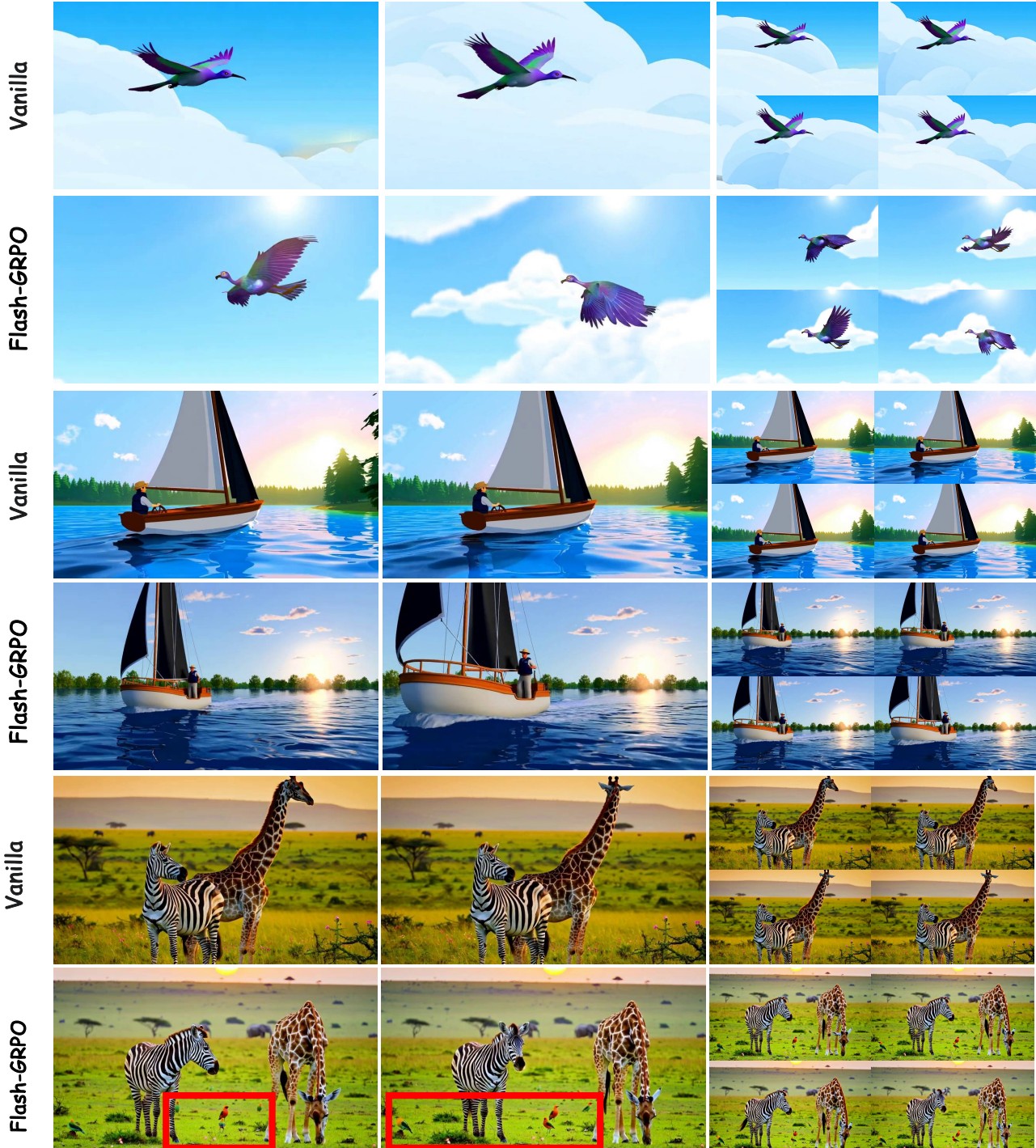

*Figure 11.* Qualitative comparison between Flash-GRPO and Vanilla with HPSv3 rewards on VBench prompts (Wan14B).

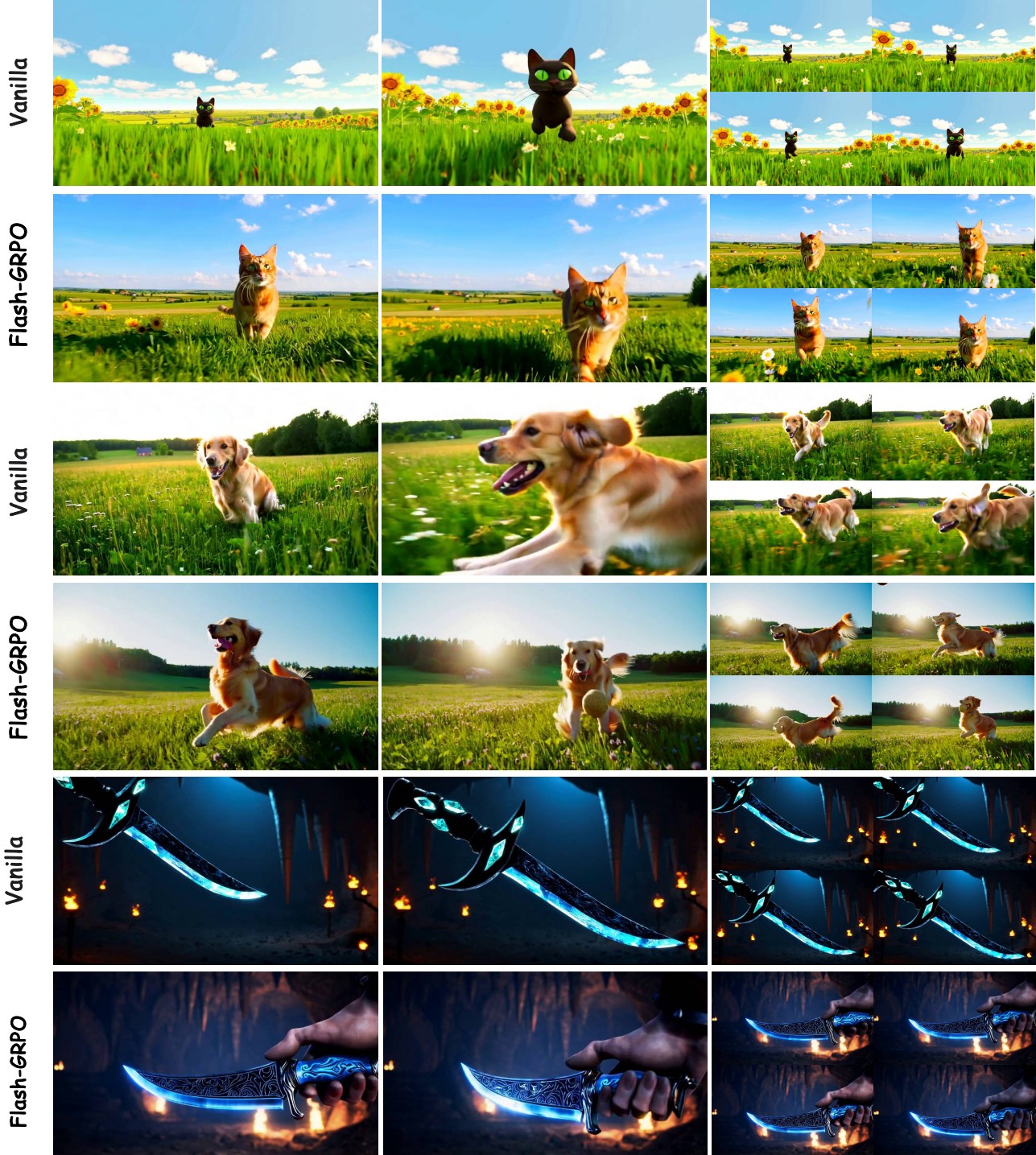

*Figure 12.* Qualitative comparison between Flash-GRPO and Vanilla with HPSv3 rewards on VBench prompts (Wan14B).

