# OpenReview forum: "Flash-GRPO: Efficient Alignment for Video Diffusion via One-Step Policy Optimization"
_ICML.cc/2026/Conference — ICML 2026 regular_

### Official Review · Reviewer_ZxT8 · 2026-03-05

**Soundness:** 3
**Presentation:** 3
**Significance:** 3
**Originality:** 3
**Overall Recommendation:** 4
**Confidence:** 4

**Summary:**

This paper proposes Flash-GRPO,  a principled single-step training framework that combines iso-temporal grouping for pre- cise advantage estimation with temporal gradient rectifi- cation for balanced optimization, achieving full trajectory performance at minimal computational cost.

**Compliance With Llm Reviewing Policy:**

Affirmed.

**Final Justification:**

The authors have provided a thorough and well-executed rebuttal, addressing all major concerns with additional experiments and clear conceptual analysis. The evidence consistently supports their claims, particularly regarding the limitations of batch-shared t and the effectiveness of Flash-GRPO. I also appreciate the significant effort invested in conducting resource-intensive experiments within a short rebuttal period. Overall, the revisions substantially strengthen the paper, and I raise my score to weak accept.

**Key Questions For Authors:**

The analysis of time-dependent scaling factors is interesting. But if you use the same t for a group, it will also avoid the problem. Can you compare the result of this setting with Temporal Gradient Rectification?

**Limitations:**

Yes.

**Strengths And Weaknesses:**

Strengths
1. The workflow of the method is presented clearly.
2. Both theoretical derivations and empirical validation support the method.

Weaknesses
1. For iso-temporal grouping, it seems just randomly select one step? What's the novelty of Sec 4.1?
2. The details of reward model are missing. In Eq.5, you mentioned process reward. Do you have different reward for different timestep under the same prompt?
3. You claimed the reward is unstable without iso-temporal grouping. Can you provide the variance of reward? The ablation studies only contain the final eval reward.

---

> ### Author Rebuttal · Authors · 2026-03-31
>
> ## W1. The novelty of Sec 4.1.
>
> **Response:** We would like to clarify an important distinction: **the novelty of Iso-Temporal Grouping (ITG) is not randomly selecting one timestep, but enforcing that all rollouts within a GRPO group share the same timestep to alleviate timestep-confounded reward variance.**
>
> Consider two designs:
>
> | Design | Timestep Assignment | Adv Reflections |
> |---|---|---|
> | Naive single-step | Each rollout gets an **independent** random timestep | Policy quality, timestep-induced variation (confounded) |
> | ITG | All rollouts in a group share the **same** timestep | Policy quality (unconfounded) |
>
> As discussed in Sec. 4.1, in the naive design (Flow-GRPO-Fast), different rollouts within the same prompt undergo SDE exploration at different timesteps. The group baseline mixes policy-induced variation with timestep-induced variation, making the advantage estimate noisy and unreliable.
>
> ITG eliminates this confounding by ensuring that all rollouts within a group share the same timestep. Different prompt groups are still assigned different timesteps, preserving temporal diversity at the global batch level.
>
> This is not a trivial design choice — as shown in Tab. 2, **naive single-step GRPO fails** (eval reward 4.64, lower than the baseline 4.67), while **adding ITG alone raises eval reward to 5.31**, a substantial improvement that demonstrates the importance of unconfounded advantage estimation.
>
> ---
>
> ## W2. The details of reward model are missing.
>
> **Response:** We would like to clarify that **there is no process reward in our formulation**. In Eq. 5, $R(x_0(x_{t_i}), c)$ denotes the reward computed on the **$x_0$**. This is consistent with Flow-GRPO / DanceGRPO (the reward is always on $x_0$).
>
> What Eq. 5 highlights is a naive solution: different rollouts undergo SDE exploration at different timesteps $t_i$. This observation is the motivation for iso-temporal grouping: if different rollouts within the same group use different $t_i$, the within-group reward variation conflates policy quality with the effect of noise injection position. ITG eliminates this confounding by forcing all rollouts to share the same timestep.
>
> Furthermore, our one-step can be viewed as an **implicit form of process-level credit assignment** — In a group, the reward signal is naturally focused on the step where exploration actually occurs, without requiring an explicit per-timestep reward model.
>
> ---
>
> ## W3. You claimed the reward is unstable without iso-temporal grouping. Can you provide the variance of reward?
>
> **Response**: Thank you for pointing this out. More precisely, without iso-temporal grouping, the reward signal used for group-relative comparison becomes timestep-confounded; we did not intend to claim that the reward itself is unstable.
>
> In naive single-step grouping, samples from different timesteps are mixed within the same group, while different timesteps have different reward statistics. This can wash out discriminative differences for timesteps with relatively small reward spread, making the resulting advantage estimates less reliable. We measured the timestep-conditional reward variance on Wan-1.3B over 50 prompts. The reward variance changes substantially across timesteps, **decreasing from 1.80 at t=0 to 0.70 at t=9**. This supports our claim that reward statistics are timestep-dependent.
>
> **The actual optimization instability is a separate issue and is addressed by TGR**. As shown in Sec. 4.2 and Fig. 8, the instability comes from the time-dependent scaling factor $\lambda(t)$, **which causes gradient norm spikes and reward collapse.** TGR corrects this imbalance.
>
> This distinction is consistent with our ablation results in Tab. 2: ITG alone already improves evaluation reward (4.64 to 5.31). Adding TGR further improves reward (to 5.42) and yields stable training.
>
> ---
>
> ## Q1. Use the same t for a group.
>
> **Response:** Thank you for this insightful question. To provide a thorough comparison, we distinguish these settings:
>
> | Setting | Timestep Assignment | Temporal Diversity | Within-Group Confounding | Cross-Group Gradient Imbalance |
> |---|---|---|---|---|
> | Naive random-t | Per-sample random t | ✓ High | ✗ Confounded | ✗ Severe |
> | Batch-shared t | One t for entire batch | ✗ None | ✓ Eliminated | ✓ Eliminated (only one t) |
> | ITG (Ours) | One t per group, different across groups | ✓ Preserved | ✓ Eliminated | ✗ Still present |
> | ITG + TGR (Ours) | ITG + gradient rectification | ✓ Preserved | ✓ Eliminated | ✓ Eliminated |
>
> We experimentally compared all settings (same training hours):
>
> | Setting | Eval |
> |---|---:|
> | Naive random-t | 4.64 |
> | Batch-shared t | 4.72 |
> | ITG | 5.31 |
> | ITG + TGR (Full) | 5.42 |
>
> **Batch-shared t can be seen as an overcorrection** — it solves the gradient imbalance problem but at the cost of temporal diversity. Our ITG + TGR combination achieves the best of both worlds: unconfounded advantage estimation, balanced gradient scales, and full temporal diversity.

---

> > ### Author Rebuttal · Reviewer_ZxT8 · 2026-04-03
> >
> > Thank you for the clarification regarding Iso-Temporal Grouping (ITG). I understand that the key idea is to enforce a shared timestep within each GRPO group.
> >
> > However, I still have some concerns regarding the level of technical novelty and overall impact of this design. In particular, for Section 4.2, I am wondering whether a simpler alternative could achieve a similar effect: for example, using the same timestep (t) across different prompts within the same batch. Intuitively, this might also mitigate timestep-related issues.
> >
> > Given this, I remain unconvinced that the proposed approach provides a fundamentally stronger solution.

---

> > > ### Author Response · Authors · 2026-04-04
> > >
> > > Thank you for the continued engagement and for acknowledging our clarification on ITG. We appreciate the opportunity to further address your concern.
> > >
> > > We believe there may be a **misunderstanding** regarding the role of batch-shared t versus our proposed framework. You suggest that using the same timestep across all prompts in a batch could achieve a similar effect. We provide both quantitative evidence and a conceptual clarification below.
> > >
> > > ## Quantitative Evidence
> > >
> > > In our first-round response Q1, we showed that **batch-shared t only achieves 4.72 on HPSv3, while Flash-GRPO achieves 5.42.** To further strengthen this comparison, we now provide a full VBench evaluation:
> > >
> > > | Method | GPU Hours | Aesthetic Quality ↑ | Imaging Quality ↑ | Subject Consistency ↑ | Object Class ↑ |
> > > |---|---:|---:|---:|---:|---:|
> > > | Wan2.1-T2V-1.3B | — | 65.46 | 66.79 | 97.56 | 88.84 |
> > > | Flow-GRPO-Fast1 | 350 | 65.92 | 65.96 | 98.46 | 88.15 |
> > > | Flow-GRPO | 350 | 65.79 | 68.60 | 97.28 | 87.92 |
> > > | Batch-shared t | 350 | 65.96 | 66.23 | 97.85 | 87.46 |
> > > | **Flash-GRPO** | **350** | **66.43** | **68.28** | **98.70** | **90.00** |
> > >
> > > Under the same 350 GPU hours budget, **Flash-GRPO achieves the best scores across all four dimensions， and  outperforms batch-shared t across all four VBench dimensions by a large margin**.
> > >
> > > ## Why Batch-Shared t Fails
> > >
> > > The reason is fundamental, not incidental. **Batch-shared t forces every prompt in the same batch to optimize at a single timestep**, **completely eliminating temporal diversity from each parameter update**. Standard diffusion training, including both pretraining and RL, relies on each sample being optimized at a different timestep to ensure the vector field is learned across the full trajectory. Batch-shared t breaks this principle: the model sees only one point on the trajectory per update, leading to extremely slow and ineffective learning (https://anonymous.4open.science/r/Flash-GRPO). It does not solve the underlying optimization challenges; it sidesteps them by crippling the training signal.
> > >
> > > ## Flash-GRPO Is a Principled Decomposition, Not a Stack of Tricks
> > >
> > > We would like to respectfully clarify that Flash-GRPO is not two independent techniques combined together, but a principled decomposition of the single-step GRPO problem into two orthogonal components:
> > >
> > > - **ITG solves the intra-group problem**: it eliminates timestep-confounded advantage estimation while preserving temporal diversity across groups.
> > > - **TGR solves the inter-group problem**: precisely because ITG preserves diverse timesteps across groups, different groups contribute gradients at vastly different scales due to λ(t). TGR normalizes this imbalance.
> > >
> > > Overall, ITG and TGR are not independent tricks; ITG preserves exactly the cross-group temporal diversity that TGR needs to operate on. Batch-shared t eliminates the need for TGR by removing temporal diversity altogether, but as the VBench results above demonstrate, this comes at a severe cost to learning effectiveness. Our framework resolves both challenges without sacrificing any desirable property.
> > >
> > > We also note that Reviewer n61K positively highlighted the elegance of our design, commenting that the proposed solutions "require virtually no additional architectural overhead."
> > >
> > > ## Summary
> > >
> > > We have now provided (1) HPSv3 reward comparison, (2) full VBench evaluation across four independent metrics, and (3) detailed conceptual analysis, all consistently demonstrating that batch-shared t is not a viable alternative to Flash-GRPO.
> > >
> > > Thank you again for the time and effort dedicated to reviewing our work. We would like to note that video RL alignment is an extremely resource-intensive research area: each experiment on 14B models requires hundreds of GPU hours, and even the 1.3B experiments in our rebuttal consumed significant computational resources. Despite these challenges, we have provided extensive additional experiments across multiple settings, baselines, and evaluation dimensions to address every concern raised. We hope that our work, which democratizes video RL alignment by making it practical and accessible to the broader community, represents a meaningful contribution to this field. **If you find that our responses have adequately addressed your concerns, we sincerely hope you might consider raising the score. Your recognition would mean a great deal to us.**

---

### Official Review · Reviewer_ycz4 · 2026-03-12

**Soundness:** 3
**Presentation:** 3
**Significance:** 3
**Originality:** 3
**Overall Recommendation:** 4
**Confidence:** 3

**Summary:**

The manuscript presents Flash-GRPO, a highly efficient single-step reinforcement learning framework designed to align video diffusion models with human preferences. The authors identify that standard full-trajectory Group Relative Policy Optimization (GRPO) suffers from prohibitive computational costs. Conversely, naive single-step subsampling leads to severe optimization instability. To bridge this gap, the paper identifies two root causes of instability: timestep-confounded variance in advantage estimation and time-dependent gradient scale imbalances. The proposed Flash-GRPO framework introduces Iso-Temporal Grouping (ITG) to ensure all rollouts for a given prompt share the same timestep, thereby isolating policy-induced variance. Furthermore, it introduces Temporal Gradient Rectification (TGR) to normalize the time-dependent scaling factor $\lambda(t)$ inherent in the SDE discretization, ensuring uniform gradient contributions across the diffusion trajectory. Empirical evaluations on Wan2.1 models (1.3B and 14B parameters) demonstrate that Flash-GRPO achieves stable monotonic reward growth and matches or exceeds full-trajectory performance with a 6x reduction in training cost.

**Compliance With Llm Reviewing Policy:**

Affirmed.

**Final Justification:**

I thank the authors for the detailed response. They fully solved my concerns. I will maintain my score, but I recommend accepting this paper.

**Key Questions For Authors:**

1. Regarding the extremely tight clip ratio of 0.001: Is this strict clipping a fundamental requirement for the convergence of the single-step SDE transition, or is it a conservative choice? How does the result behave if this ratio is relaxed to standard values like 0.1 or 0.2?

2. The derivation of $\lambda(t)$ assumes the Gaussian transition kernel induced by the Euler-Maruyama discretization. How well does Temporal Gradient Rectification generalize if one were to use a different, perhaps higher-order, SDE solver for the exploration step?

3. In Figure 4, Flow-GRPO-Fast1 experiences a catastrophic reward collapse. Even with TGR rectifying the gradient magnitudes, how does Flash-GRPO prevent catastrophic forgetting of the original deterministic ODE trajectory when only a single step is optimized per rollout?

4. Could the authors provide a comparative analysis between Flash-GRPO and offline preference optimization methods?

5. Why was Dance-GRPO excluded from the baseline comparisons in Table 1 despite being heavily referenced and its dataset directly utilized in the experimental setup?

**Limitations:**

yes

**Strengths And Weaknesses:**

## Strength

1. The paper addresses a critical scalability bottleneck in generative AI. Reducing the alignment cost of a 14B parameter video model by a factor of 6 while maintaining stability is a highly valuable contribution to the community.

2. The derivation of the time-dependent scaling factor $\lambda(t)$ in Equation 10 explicitly identifying how the Euler-Maruyama discretization injects magnitude imbalance into the policy gradient, the authors provide a rigorous mathematical justification for their TGR module rather than relying on heuristic gradient clipping.

3. Iso-Temporal Grouping is a remarkably simple yet conceptually sound approach to decoupling policy performance from timestep difficulty. It elegantly solves the confounding variable problem in advantage estimation without introducing additional computational overhead.

## Weakness

1. Single-step optimization methods can sometimes be prone to reward hacking, where the model learns to game the specific reward model (e.g., HPSv3) at the expense of overall distribution fidelity. While the paper evaluates VBench metrics, a deeper discussion on whether single-step updates exacerbate reward over-optimization compared to full-trajectory updates would strengthen the analysis.

2. The implementation details mention a very strict GRPO clip ratio of 0.001 to ensure stable policy updates under the single-step paradigm. This is significantly smaller than standard PPO/GRPO implementations. The paper lacks an analysis of how sensitive the Flash-GRPO framework is to this specific hyperparameter.

3. While ITG perfectly isolates temporal variance within a prompt group, it inherently restricts the temporal diversity within a single parameter update step for that specific group. The global batch maintains diversity, but it is worth analyzing if this localized temporal homogeneity affects the learning dynamics of the vector field $v_\theta(x_t, t)$ over long training horizons.

---

> ### Author Rebuttal · Authors · 2026-03-31
>
> ## W1. Single-step optimization may be prone to reward hacking.
>
> **Response:**
>
> **Theoretically, across the global batch, each unique prompt group is assigned a distinct timestep, ensuring that the optimization covers diverse timesteps throughout training**.
>
> **Cross-reward evaluation.** We train with HPSv3 and eval on Motion Quality (VideoAlign). Flash-GRPO achieves comparable Motion Quality relative to Flow-GRPO. **This cross-reward protocol confirms that our method does not introduce additional reward hacking**.
>
> | Method | HPSv3 | Motion Quality |
> |---|---:|---:|
> | Flow-GRPO | 5.14 | -0.445 |
> | **Flash-GRPO** | 5.42 | -0.443  |
>
> **VBench.** Tab. 1 provides further evidence: Flash-GRPO achieves the highest Aesthetic Quality and Subject Consistency. **These independent metrics show no degradation, confirming that Flash-GRPO is not hacking.**
>
> ---
>
> ## W2 (Q1). The clip ratio sensitivity.
>
> **Response:** **This is not a specific choice; Flow-GRPO and DanceGRPO also adopt 1e-3 to 1e-4.** The LLM-standard values of 0.1–0.2 are not directly transferable to diffusion RL. We provide a sensitivity analysis:
>
> | Clip Ratio | Eval Reward (HPSv3) |
> |---|---:|
> | Flow-GRPO-Fast1 (0.001) | 4.64 |
> | 0.001 (default) | 5.42 |
> | 0.1 | 5.29 |
> | 0.2 | 4.75 |
>
> Larger clip ratios lead to lower final reward, but Flash-GRPO outperforms Flow-GRPO-Fast1 at all tested values, confirming the gains stem from ITG and TGR rather than a specific clipping choice.
>
> ---
>
> ## W3. The long-horizon learning dynamics.
>
> **Response:** ITG does not remove temporal diversity; it **reorganizes** where diversity is introduced. In naive single-step GRPO, diversity is injected within each group by assigning different timesteps to different rollouts, which is precisely the source of the confounding problem identified in Sec. 4.1. ITG eliminates this confounding while preserving temporal diversity at the **global batch level**, where different prompt groups are assigned different timesteps. The vector field continues to optimize all timesteps across groups.
>
> Meanwhile, we do not observe evidence that this localized temporal homogeneity harms long-horizon learning. On the contrary, Fig. 4 and 5 show that Flash-GRPO maintains stable reward growth. Additionally, VBench results in Tab. 1, which evaluate the full generation pipeline across all timesteps, show consistent improvements over both baselines.
>
> ---
>
> ## Q2. Different SDE.
>
> **Response:** We focus on Euler because it is the standard solver used in diffusion RL (Flow-GRPO, DanceGRPO). However, **TGR is solver-agnostic**.
>
> Additionally, we tested an different $\sigma_t$ of $\lambda(t)$:
>
> | Diffusion Term | Method | Eval HPSv3 |
> |---|---|---:|
> | $\sigma_t = t$ (default) | Flow-GRPO-Fast1 | 4.65 |
> | $\sigma_t = t$ (default) | Flash-GRPO | 5.42 |
> | $\sigma_t = \sqrt{t}$ | Flow-GRPO-Fast1 | 4.99 |
> | $\sigma_t = \sqrt{t}$ | Flash-GRPO | 5.41 |
>
> Flash-GRPO maintains consistent gains. We view extending TGR to higher-order solvers as straightforward future work.
>
> ---
>
> ## Q3. Catastrophic forgetting.
>
> **Response:** We would like to first clarify that the collapse of Flow-GRPO-Fast1 (Fig. 4) is not caused by forgetting, but by timestep-confounded advantage estimation and imbalanced gradient scales.
>
> As for the concern of whether single-step optimization degrades the model's capabilities at non-optimized timesteps, we note that **diffusion SFT follows single-step paradigm** (each sample computes loss at one randomly sampled timestep). Additionally, timestep sampling ensures uniform coverage across iterations. Empirically, the VBench metrics in Tab. 1, which evaluate generation quality requiring all timesteps works correctly.
>
> ---
>
> ## Q4. Comparative analysis with offline preference optimization methods.
>
> **Response:** **Practically**, there is no established open-source video diffusion offline preference optimization baseline comparable to our setting. Existing methods (e.g., Diffusion-DPO) are primarily designed for image generation. Moreover, online RL alignment has an important practical advantage for video: it does **not require offline preference pairs**.
>
> **Additionally**, the motivation of our work is to address the challenge in **video GRPO**: timestep confounding and gradient imbalance. These do not exist in offline methods. Flash-GRPO is designed to solve these two issues. A direct comparison with offline methods would conflate algorithmic efficiency with the online-vs-offline paradigm choice, which is an orthogonal research question.
>
> ---
>
> ## Q5. DanceGRPO Analysis.
>
> **Response:** DanceGRPO and Flow-GRPO are very similar. We chose Flow-GRPO as the baseline because it provides fast variants, enabling a direct comparison across efficiency.
>
> We additionally eval DanceGRPO:
>
> | Method | Eval HPSv3 |
> |---|---:|
> | DanceGRPO |  5.06 |
> | Flow-GRPO | 5.14 |
> | Flash-GRPO | 5.42 |
>
> The results confirm that DanceGRPO performs comparably to Flow-GRPO under matched settings, and Flash-GRPO outperforms both methods.

---

> > ### Author Rebuttal · Reviewer_ycz4 · 2026-04-03
> >
> > I appreciate the authors’ rebuttal, which addresses my concerns. I will maintain my original score.

---

> > > ### Author Response · Authors · 2026-04-04
> > >
> > > Thank you for acknowledging that our responses have **adequately addressed** your concerns. We are grateful for your constructive feedback throughout this process.
> > >
> > > We believe this work makes a meaningful contribution to the video generation community: by identifying and resolving the **two root causes of single-step GRPO instability**, Flash-GRPO brings the training cost of video RL alignment down to a level that is **practical for standard academic labs**. In this rebuttal, we have provided cross-reward evaluation, clip ratio ablation, DanceGRPO comparison, and alternative SDE experiments, which we believe **comprehensively address every concern raised in the review**. **We would be truly grateful if this additional effort could be reflected in the final score, and we deeply appreciate your engagement with our work.**

---

### Official Review · Reviewer_n61K · 2026-03-12

**Soundness:** 3
**Presentation:** 3
**Significance:** 2
**Originality:** 3
**Overall Recommendation:** 4
**Confidence:** 4

**Summary:**

This paper introduces Flash-GRPO, an efficient single-step training framework designed to alleviate the massive computational bottleneck of applying Group Relative Policy Optimization (GRPO) to large-scale video diffusion models. The authors identify that naive single-step subsampling in GRPO leads to severe optimization instability due to two primary factors: timestep-confounded advantage estimation and gradient scale heterogeneity across the diffusion trajectory. To resolve these, the paper proposes (1) Iso-temporal grouping, which forces all rollouts in a GRPO group to share the same timestep, isolating policy variance from timestep difficulty, and (2) Temporal gradient rectification, which explicitly neutralizes the time-dependent scaling factor ($\lambda(t)$) inherent in the diffusion SDE. The method achieves a 6x training acceleration compared to full-trajectory baselines and is successfully validated on massive foundational models (up to 14B parameters), achieving state-of-the-art alignment quality on benchmarks.

**Compliance With Llm Reviewing Policy:**

Affirmed.

**Final Justification:**

The authors’ rebuttal fully resolved my concerns. In particular, the additional results on T2I, reward generalization, and timestep diversity were helpful in clarifying both the scope and the practical value of the method.

**Key Questions For Authors:**

- While the paper focuses heavily on the computational bottlenecks of large-scale video diffusion models, does the Flash-GRPO framework (specifically iso-temporal grouping and temporal gradient rectification) seamlessly transfer to Text-to-Image (T2I) models like Stable Diffusion? Given that T2I models also suffer from alignment inefficiencies, do you have any preliminary empirical results or theoretical reasons to expect similar acceleration and performance gains in the image domain? Moreover, do authors think their method can be directly applied to other policy gradient methods or DPO in diffusion models? If so, would Flash-DPO be possible from this framework? I know it's too much to ask, but it would be interesting to see if the method could be applied to various senario (supervised vs. reward-based), loss functional, and tasks (T2I, T2V).

- In iso-temporal grouping, all samples for a specific prompt are evaluated at the exact same timestep. Does this reduce the temporal diversity of the gradient updates per batch, and did you observe any need to increase batch sizes to ensure the model sees a sufficient distribution of timesteps simultaneously?

- In training phase, have authors tried to sample $x_0$ by using various diffusion term when generating with SDEs?

**Limitations:**

Yes.

**Strengths And Weaknesses:**

**Strength**

- The proposed solutions—iso-temporal grouping and temporal gradient rectification—are highly elegant. They require virtually no additional architectural overhead or complex hyperparameter tuning, making them easily adoptable by the broader community.

- Validating an RL alignment method on a 14-billion-parameter video generation model (Wan14B) is a highly commendable engineering feat. Demonstrating a 6x speedup while simultaneously outperforming full-trajectory baselines (Flow-GRPO) in Aesthetic Quality and Subject Consistency provides undeniable proof of the method's practical utility.

**Weakness**

- The proposed framework is tightly coupled to the GRPO algorithm. It is not immediately clear whether the core insights, particularly iso-temporal grouping, naturally extend to other widely adopted alignment algorithms for diffusion models, such as Direct Preference Optimization (DPO). Because offline algorithms like DPO inherently structure their sample comparisons differently, the inability to generalize these specific efficiency gains to other paradigms may limit the broader theoretical impact and adoption of the work.

- While the results on large-scale video diffusion models are impressive, the empirical evaluation is somewhat restricted in scope. To convincingly establish Flash-GRPO as a general-purpose efficiency framework for generative alignment, the study would significantly benefit from demonstrating consistent trends across a broader variety of settings. Specifically, the paper lacks validation on different generative modalities (e.g., Text-to-Image [T2I] models), diverse reward models, and alternative training scenarios (such as Supervised Fine-Tuning [SFT]).

---

> ### Author Rebuttal · Authors · 2026-03-31
>
> ## W1. The proposed framework is tightly coupled to the GRPO algorithm.
>
> **Response:** Thank you for this thoughtful comment.
>
> The key observation is that **the efficiency problem Flash-GRPO solves is specific to online RL alignment, not to SFT or DPO.** SFT and DPO already operate in a single-step paradigm. The online GRPO setting introduces two *new* difficulties absent from single-step objectives: (1) credit assignment, where mixing timesteps within a group; and (2) timestep-dependent gradient imbalance from SDE. Our contribution is making single-step GRPO as efficient as SFT-style optimization.
>
> Beyond the above difficulties, GRPO is also the most practical alignment paradigm for video generation. Collecting DPO pairs is harder than for images, since judgments must jointly consider motion quality, visual quality, and prompt alignment. Online RL alignment is therefore practical for video, but it is computationally expensive.
>
> While our work focuses on GRPO, the insights are not algorithm-specific. **Temporal gradient rectification (TGR) addresses a property of the SDE**: the scaling factor $\lambda(t)$ exists in any policy gradient computed through the SDE, regardless of whether the algorithm is GRPO, PPO, or REINFORCE. Similarly, the **iso-temporal grouping (avoiding timestep-confounded comparisons) could inform DPO-style methods** through timestep-matched pair construction. We view such extensions as promising future work.
>
> ---
>
> ## W2 (Q1). Broader empirical validation across modalities, reward models, and training scenarios.
>
> **Response:** Thank you for this suggestion.
>
> **T2I.** We evaluate Flash-GRPO using Qwen-Image (512 px) with PickScore. Under the same training budget, Flash-GRPO achieves the highest PickScore, confirming that the efficiency gains in image domain.
>
> | Method | PickScore |
> |---|---:|
> | Qwen-Image | 22.2 |
> | Flow-GRPO-Fast1 | 22.8 |
> | **Flash-GRPO** | **23.2** |
>
> **Reward generalization.** The main paper already demonstrates generalization across rewards: we report results with **HPSv3** in Tab 1–2 and Fig 4–5, and **Motion Quality** in Fig 6. To further address this concern, we additionally evaluate on **VideoAlign VQ** as a third independent reward:
>
> | Reward | Method | Eval |
> |---|---|---:|
> | HPSv3 | Flow-GRPO-Fast1 | 4.64 |
> | HPSv3 | **Flash-GRPO** | **5.42** |
> | MQ | Flow-GRPO-Fast1 | -0.34 |
> | MQ | **Flash-GRPO** | **-0.28** |
> | VQ | Flow-GRPO-Fast1 | -0.56 |
> | VQ | **Flash-GRPO** | **-0.46** |
>
> Flash-GRPO shows consistent improvements across all three rewards, confirming that its effectiveness is not tied to a single reward signal.
>
> **SFT.** We appreciate this suggestion but would like to clarify that SFT and GRPO address different efficiency regimes. SFT already operates as single-step optimization, so the bottleneck Flash-GRPO resolves does not arise. In this sense, **Flash-GRPO's contribution is precisely to bridge the efficiency gap between RL and SFT**, making the former as cheap as the latter.
>
> Overall, we would like to emphasize the practical impact: with Flash-GRPO, **Wan2.1-1.3B (480p, 5s)** can be post-trained on a single **8-GPU machine in a little over one day**, while achieving significant gains on VBench.
>
> ---
>
> ## Q2. Does iso-temporal grouping reduce temporal diversity? Is a larger batch size needed?
>
> **Response:** **Iso-temporal grouping enforces temporal homogeneity within each prompt group, but temporal diversity is fully preserved at the global batch level**.
>
> In our framework, batch size = num groups × group size (8). More groups means more distinct timesteps are covered. For Wan2.1-1.3B, our default setting uses 48 prompt groups.
>
> To directly test the effect of temporal coverage, we varied the number of prompt groups (batch size) while keeping the group size fixed:
>
> | Batch Size (Num Groups) | HPSv3 |
> |---|---|
> | Flow-GRPO-Fast1 (48)|4.64|
> | Flash-GRPO (12) | 4.91 |
> | Flash-GRPO (24) | 5.24 |
> | Flash-GRPO (48) | 5.42 |
>
> We observe that increasing the number of groups (broader timestep coverage per update) consistently improves performance, which confirms broader timestep coverage provides additional gains. Meanwhile, the method does not require unusually large batch sizes.
>
> ---
>
> ## Q3. Various diffusion terms in SDE.
>
> **Response:** Thank you for this question.
>
> **Theoretically**, TGR is robust to the choice of $\sigma_t$ in Eq. 4. **Additionally**, we tested an alternative diffusion term to validate this robustness:
>
> | Diffusion Term | Method | Eval Reward (HPSv3) |
> |---|---|---:|
> | $\sigma_t = t$ (default) | Flow-GRPO-Fast1 | 4.65 |
> | $\sigma_t = t$ (default) | Flash-GRPO | 5.42 |
> | $\sigma_t = \sqrt{t}$ | Flow-GRPO-Fast1 | 4.99 |
> | $\sigma_t = \sqrt{t}$ | Flash-GRPO | 5.41 |
>
> Flash-GRPO maintains consistent gains under both diffusion terms, confirming that TGR generalizes across different SDE parameterizations. The performance of Flash-GRPO itself is nearly identical (5.42 vs 5.41), demonstrating strong robustness to the choice of noise schedule.

---

> > ### Author Rebuttal · Reviewer_n61K · 2026-04-03
> >
> > The authors’ rebuttal fully resolved my concerns. In particular, the additional results on T2I, reward generalization, and timestep diversity were helpful in clarifying both the scope and the practical value of the method. I'll maintain the score.

---

> > > ### Author Response · Authors · 2026-04-04
> > >
> > > Thank you for confirming that our responses have **fully resolved** your concerns. We truly appreciate the time you spent evaluating our additional experiments.
> > >
> > > We believe video RL alignment is an important yet under-explored problem: training a 14B video model with GRPO currently requires hundreds of GPU hours, which severely limits both research iteration and practical deployment. Flash-GRPO reduces this cost by **6× while maintaining or exceeding full-trajectory performance**, making video alignment accessible to the broader research community. In this rebuttal, we have further provided T2I experiments, three different reward models (HPSv3, Motion Quality, VideoAlign VQ), batch size analysis, and alternative diffusion term validation. **We sincerely hope these efforts might be reflected in the final score, and your support would mean a great deal to us.**

---

### Decision · Program_Chairs · 2026-04-30

**Decision:**

Accept (regular)

**Comment:**

The paper proposes Flash-GRPO, an efficient single-step GRPO framework for aligning large-scale video diffusion models.
Reviewers agreed that the work addresses an important computational bottleneck and that the proposed iso-temporal grouping and temporal gradient rectification are simple, well motivated, and practically useful. The method is validated on large video diffusion models, including Wan2.1-1.3B and 14B, and shows substantial training acceleration while matching or outperforming full-trajectory GRPO baselines. The rebuttal further strengthens the paper with additional T2I results, multiple reward-model evaluations, clip-ratio sensitivity analysis, alternative SDE tests, DanceGRPO comparison, and batch-shared timestep baselines. Remaining limitations include the focus on GRPO-style online RL and the need for broader validation across more solvers, tasks, and preference-optimization paradigms.
However, these limitations mainly affect scope rather than the validity or impact of the contribution. Overall, AC recommends acceptance, as the paper offers a timely, technically sound, and practically impactful solution for scalable video diffusion alignment.